# HDAC1 and HDAC2 integrate checkpoint kinase phosphorylation and cell fate through the phosphatase-2A subunit PR130

Anja Göder[1], Claudia Emmerich[2], Teodora Nikolova[1], Nicole Kiweler[1], Maria Schreiber[2], Toni Kühl[3], Diana Imhof [3], Markus Christmann[1], Thorsten Heinzel[2], Günter Schneider[4] & Oliver H. Krämer[1]

Checkpoint kinases sense replicative stress to prevent DNA damage. Here we show that the histone deacetylases HDAC1/HDAC2 sustain the phosphorylation of the checkpoint kinases ATM, CHK1 and CHK2, activity of the cell cycle gatekeeper kinases WEE1 and CDK1, and induction of the tumour suppressor p53 in response to stalled DNA replication. Consequently, HDAC inhibition upon replicative stress promotes mitotic catastrophe. Mechanistically, HDAC1 and HDAC2 suppress the expression of PPP2R3A/PR130, a regulatory subunit of the trimeric serine/threonine phosphatase 2 (PP2A). Genetic elimination of PR130 reveals that PR130 promotes dephosphorylation of ATM by PP2A. Moreover, the ablation of PR130 slows G1/S phase transition and increases the levels of phosphorylated CHK1, replication protein A foci and DNA damage upon replicative stress. Accordingly, stressed PR130 null cells are very susceptible to HDAC inhibition, which abrogates the S phase checkpoint, induces apoptosis and reduces the homologous recombination protein RAD51. Thus, PR130 controls cell fate decisions upon replicative stress.

[1] Institute of Toxicology, University Medical Center Mainz, Obere Zahlbacher Strasse 67, 55131 Mainz, Germany. [2] University of Jena, Institute of Biochemistry and Biophysics, Center for Molecular Biomedicine (CMB), Hans-Knöll-Strasse 2, 07745 Jena, Germany. [3] Pharmaceutical Biochemistry and Bioanalytics, Pharmaceutical Institute, University of Bonn, An der Immenburg 4, 53121 Bonn, Germany. [4] Klinik und Poliklinik für Innere Medizin II, Technical University of Munich, Ismaningerstrasse 22, 81675 Munich, Germany. Anja Göder and Claudia Emmerich contributed equally to this work Correspondence and requests for materials should be addressed to O.H.Käm. (email: okraemer@uni-mainz.de)

Disturbances in the progression of DNA replication forks and DNA damage activate checkpoint kinases, including ataxia telangiectasia mutated (ATM), ATM and Rad3-related (ATR), checkpoint kinase-1 (CHK1), and checkpoint kinase-2 (CHK2)[1–5]. These kinases promote cell cycle arrest and the intra S phase checkpoint[3–5], which prevent fatal premature transitions of cells with incompletely replicated or damaged DNA into mitosis[6,7]. Checkpoint kinases also modulate DNA repair as well as pro-apoptotic signalling to prevent genomic instability and cell transformation[3,8,9].

Obstructed replication forks activate ATR and its downstream targets CHK1 and ATM[3,10–13]. This activation of ATM and CHK1 in cells exposed to replicative stress does not require the MRE11/RAD51/Nibrin (MRN) complex that promotes autophosphorylation of ATM in response to direct DNA double-strand breaks (DSBs)[10]. Upon replicative stress, ATR activates replication protein A (RPA), which binds and protects single-stranded DNA (ssDNA)[2,12,14,15]. The homologous recombination (HR) pathway repairs collapsed replication forks and ensuing DSBs[16]. CHK1 and the WEE1/Cyclin-dependent kinase-1 (CDK1) signalling node regulate DNA replication origin firing during S phase and transition into G2/M phase by an inhibition of the CDK1/Cyclin B complex. Furthermore, WEE1 regulates the DNA replication checkpoint by its ability to control histone synthesis[7,17–19]. Checkpoint kinases also regulate cell cycle progression and cell fate through an activation of the tumour-suppressive transcription factor p53, which can induce cell cycle arrest and cell death. Accordingly, p53-negative cells rely strongly on checkpoint kinases to stall their cell cycle and to survive replicative stress[7].

Dephosphorylation is the most straightforward way to inactivate checkpoint kinases. PP2A complexes, which consist of the subunits PP2A-A (structural component, PPP2R1A/B), PP2A-B (at least 17 B subunits, provide substrate specificity) and PP2A-C (catalytic activity, PPP2CA/B)[20], target CHK1/CHK2 (refs. [21–23]). A constitutive interaction with PP2A-A and PP2A-C has been reported to prevent the phosphorylation of ATM at S1981 in human lymphoblastoid cells[24]. However, others found that inhibiting PP2A in Xenopus egg extracts has no impact on the phosphorylation of ATM[25,26] and immunoprecipitated ATM from untreated cells can phosphorylate itself and its targets in vitro[27,28]. Prominent roles of PP2A-B subunits for cell fate decisions have been identified for interleukin 2 (IL-2) deprivation-induced T-cell apoptosis, embryonic development and tumourigenesis[29–32].

The histone deacetylase (HDAC) family, which falls into four classes (I, IIa/IIb, III and IV), deacetylates lysine residues[33]. Recent observations demonstrate that the class I HDACs, HDAC1, HDAC2 and HDAC3, modulate DNA damage signalling[34–36], maintain genomic stability and prevent tumourigenesis in vivo[37–41]. Accordingly, inhibitors of HDACs (HDACi) enhance the cytotoxicity of DNA-damaging chemotherapies and of drugs targeting S phase and DNA repair[42].

It remains to be identified how HDACs modulate checkpoint kinase signalling precisely. Our data reveal that HDAC1 and HDAC2 maintain checkpoint kinase signalling, cell cycle arrest and survival through a suppression of PR130. This newly defined mechanism connects epigenetic modifiers to checkpoint kinase signalling and cell cycle progression during replicative stress.

## Results

### HDACs sustain checkpoint kinase phosphorylation.
We analysed whether class I HDACs regulate checkpoint kinase phosphorylation. We treated HCT116 and RKO colon cancer cells and murine embryonic fibroblasts (MEFs) with hydroxyurea,

ultraviolet light or 5-fluorouracil. Such agents impede the progression of replication forks and activate checkpoint kinases[5,11–13,43]. To study the impact of HDACs, we specifically inhibited HDAC1,-2,-3 with the benzamide MS-275 (ref. [44]).

Western blot analyses showed that hydroxyurea induced the phosphorylation of ATM and ATR in HCT116 cells (Fig. 1a). MS-275 significantly decreased ATM phosphorylation at S1981 after a 24-h treatment, but not ATR phosphorylation at T1989 (Figs. 1a, b). MS-275 additionally diminished the hydroxyurea-induced phosphorylation of CHK1 at S317 and CHK2 at T68 (Figs. 1c, d). These effects of MS-275 were not due to reduced checkpoint kinase expression (Figs. 1a–d). Within 6 h, MS-275 did not alter the hydroxyurea-induced phosphorylation of ATM, CHK1 and CHK2 (Fig. 1c). Thus, MS-275 antagonises checkpoint kinase signalling in a time-delayed manner.

MS-275 also diminished the phosphorylation of ATM and CHK1/CHK2 that is induced by ultraviolet light and 5-fluorouracil, but affected ATR marginally (Fig. 1e; Supplementary Fig. 1a). Furthermore, MS-275 blocked checkpoint kinase phosphorylation in RKO cells and MEFs, and the structurally unrelated HDACi LBH589 abrogated checkpoint kinase phosphorylation in HCT116 cells (Supplementary Fig. 1b-d).

Hence, an inhibition of class I HDACs suppresses the phosphorylation of ATM, CHK1 and CHK2 in response to various sources of replicative stress and in several cellular systems.

### HDACi promotes S phase slippage upon replicative arrest.
Since activated checkpoint kinases restrict cell cycle progression, we hypothesised that MS-275 disturbs this mechanism. While hydroxyurea arrested HCT116 cells in S phase and delayed their transition into G2/M phase, MS-275 induced a cell cycle arrest in G1 phase (Figs. 2a, b; see legend for exact numbers). As expected from its ability to block checkpoint kinase phosphorylation (Fig. 1), MS-275 led to a slippage of hydroxyurea-treated cells into G2 phase (Figs. 2a, b). This was associated with an increase in the number of mitotic figures (Fig. 2c).

We additionally assessed DNA synthesis using the DNA fibre assay (Fig. 2d). Cells that were treated with hydroxyurea and MS-275 incorporated more labelled nucleotides and re-started DNA synthesis faster than hydroxyurea-treated cells (Fig. 2e). These effects should be appreciated with the notion that the incorporation time for each nucleotide was limited to 20 min after the removal of hydroxyurea and MS-275.

We then analysed WEE1 and p53 because of their well-established functions for the intra S phase checkpoint[7,18,19]. MS-275 reduced WEE1 together with the phosphorylation of its target CDK1 (Fig. 2f). The levels of Cyclin B, which promotes mitotic entry with unphosphorylated CDK1, were elevated in cells exposed to hydroxyurea and hydroxyurea plus MS-275 (Fig. 2g). Regarding p53, hydroxyurea induced and MS-275 reduced p53 levels and its checkpoint kinase-dependent phosphorylation (Figs. 2h, i). Accordingly, MS-275 attenuated the p53-dependent accumulation of the phosphatase WIP1. Hydroxyurea and MS-275 alone increased the expression of the CDK inhibitor p21, but their combination decreased p21 (Fig. 2h).

We deduce that cells exposed to hydroxyurea and MS-275 escape from the hydroxyurea-induced S phase block and traverse into G2 phase and catastrophic mitosis.

### Impact of MS-275 on apoptosis and replication fork stability.
Next, we analysed how hydroxyurea and MS-275 affected the survival of HCT116 cells. After a 48-h incubation time, hydroxyurea caused 21% apoptotic cell death. Addition of MS-275 to such cells significantly increased apoptosis to 34%

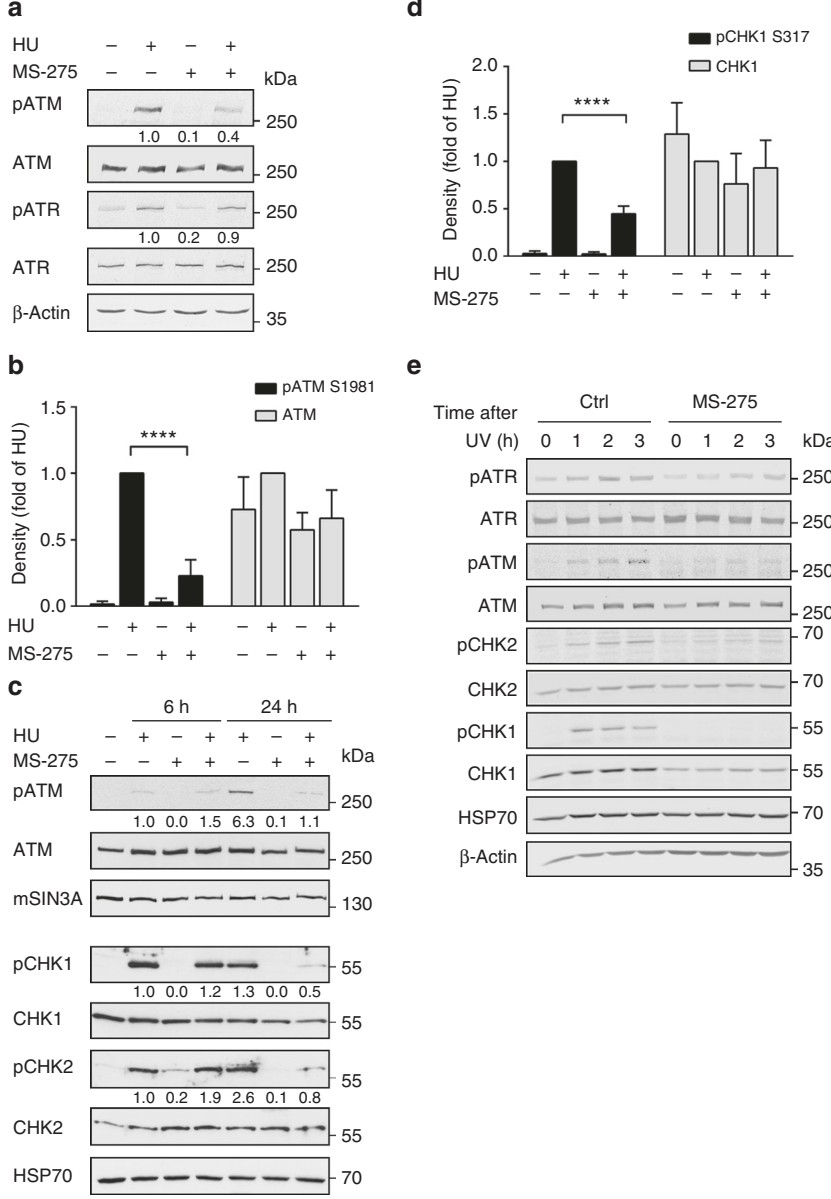

**Fig. 1** HDACi impairs checkpoint signalling upon replicative stress. **a** HCT116 cells were stimulated with 1 mM hydroxyurea (HU) and 2 μM MS-275 for 24 h. Western blot analyses of whole-cell lysates were performed to detect ATM and ATR as well as their phosphorylated forms (pATM S1981; pATR T1989). β-Actin served as loading control. Numbers indicate densitometric analysis of signals relative to HU-treated samples and normalised to β-Actin (n = 3). **b** Densitometric evaluation of ATM phosphorylation at S1981 and ATM levels (after 24 h) following protein detection via immunoblot. Data were normalised to the loading controls. The respective amounts (phosphorylated) proteins are compared to those of HU-treated cells. Results represent the mean ± SD (n = 3; one-way ANOVA; ****P < 0.0001). **c** Cells were treated with 1 mM HU and 2 μM MS-275 for 6–24 h. Whole-cell extracts were blotted to two membranes to detect phosphorylated and total levels of ATM, CHK1 (pCHK1 S317) and CHK2 (pCHK2 T68) by immunoblot. HSP70 and mSIN3A are loading controls. Numbers indicate densitometric analysis of signal relative to HU-treated cells and normalised to the respective loading control (n = 3). **d** Densitometric analysis of phosphorylated (S317) and total CHK1 levels after 24 h detected via western blot. Data were normalised to the respective loading controls. Results display relative amounts of pCHK1/CHK1 compared to HU-treated cells as mean ± SD (n = 3). Statistical analysis was performed using one-way ANOVA (****P < 0.0001). **e** Cells were either pre-incubated with 2 μM MS-275 or were left untreated for 24 h and subsequently exposed to 10 J/m² UVC. Plates were harvested 0, 1, 2 and 3 h after irradiation. Indicated proteins were detected via western blot. HSP70 and β-Actin served as loading controls

(Fig. 3a). We could confirm these results by analysing the cleavage of the caspase substrate poly(ADP-ribose)-polymerase-1 (PARP1) and by measuring subG1 fractions (Fig. 3b; Supplementary Fig. 2a). These results are consistent with previous data[45,46].

Since hydroxyurea leads to an accumulation of ssDNA stretches that are converted into DSBs after replication fork collapse[12], we tested the impact of MS-275 on the formation of ssDNA and DSBs. To detect ssDNA we measured the presence of

RPA foci. These accumulate in the S phase of cells exposed to hydroxyurea, because there is ongoing DNA helicase activity and lagging DNA polymerase activity. MS-275 significantly reduced RPA foci in hydroxyurea-treated HCT116 cells from an average of 15 foci/cell to 10 foci/cell (Figs. 3c, d), which is coherent with S phase progression (Figs. 2a–e).

To detect DSBs, we measured the levels of phosphorylated histone H2AX (γH2AX, phosphorylated at S139), which indicates collapsed replication forks in hydroxyurea-treated cells[12,16].

Confocal microscopy and immunoblotting revealed that compared to cells treated with hydroxyurea alone, cells treated with hydroxyurea plus MS-275 harboured increased amounts of γH2AX (Figs. 3e, f).

Thus, we conclude that MS-275 reduces hydroxyurea-induced RPA foci, while γH2AX foci persist.

**Relevance of checkpoint kinases in hydroxyurea-treated cells.** MS-275 blocks checkpoint kinase phosphorylation in hydroxyurea-treated cells. We tested the relevance of this finding by inhibition of checkpoint kinases.

We used KU-60019, currently the most specific inhibitor of ATM[47], and eliminated ATM with RNAi. Both strategies

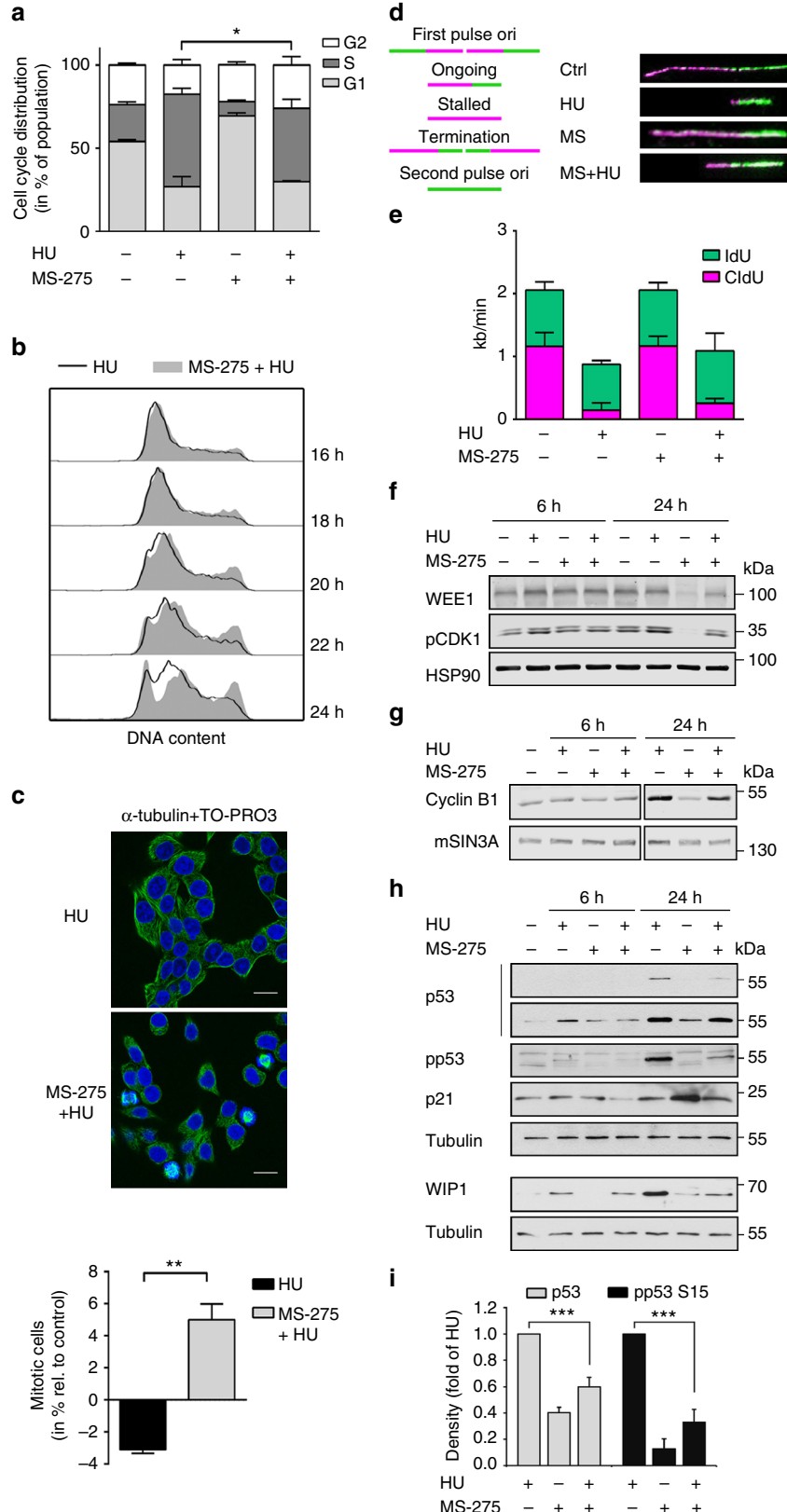

increased cell death rates in hydroxyurea-exposed HCT116 cells and ATM inhibition in the presence of hydroxyurea increased the levels of γH2AX (Fig. 4a-c).

Since ATR maintains replication fork integrity[12], we analysed its role using the specific ATR inhibitor VE-821 (ref. [48]) and RNAi. HCT116 cells treated with hydroxyurea had significantly higher rates of apoptosis and augmented DNA damage when we additionally inhibited or eliminated ATR (Figs. 4c–f).

Since ATR prevents the collapse of stalled replication forks into γH2AX-positive DSBs[12,19], the accumulation of γH2AX in cells treated with hydroxyurea and ATRi should differ from the induction of pATM by activated ATR during replicative stress[10]. The neutral comet assay detecting DSBs confirmed that hydroxyurea could induce checkpoint kinase phosphorylation independent of DSBs (Supplementary Fig. 2b; Fig. 1c). Thus, high pATM levels in hydroxyurea plus ATRi-treated cells likely stem from DSBs activating autophosphorylation of ATM.

To analyse functions of CHK1, which we verify as a bona fide downstream target of ATR in hydroxyurea-treated cells (Figs. 4d,f), we used the CHK1 inhibitors LY2603618 and MK-8776 (refs. [49,50]). These compromised the survival and DNA integrity of hydroxyurea-treated cells (Figs. 4g, h).

Next, we analysed functions of CHK2 with isogenic CHK2$^{-/-}$ HCT116 cells. Compared to wild-type cells, CHK2$^{-/-}$ cells show less PARP1 cleavage and reduced γH2AX (Fig. 4i). These data disfavour that CHK2 promotes the survival and DNA repair activity of hydroxyurea-treated HCT116 cells.

ATM, ATR and CHK1 protect HCT116 cells from apoptosis and DNA damage in response to hydroxyurea. Such findings suggest that the reduction of pATM and pCHK1 by HDACi is biologically important for cells with replicative stress.

**HDAC1 and HDAC2 repress PR130 expression**. Next, we investigated by which mechanisms HDACi disturb checkpoint kinase phosphorylation. MS-275 does not impair the hydroxyurea-induced checkpoint kinase phosphorylation within 6 h (Fig. 1c), which suggests that MS-275 induces a time-delayed expression of negative regulator(s) of checkpoint kinases. If this is the case, a pretreatment with MS-275 should affect hydroxyurea-induced phosphorylation events immediately. Indeed, a 16-h pretreatment with MS-275 antagonised checkpoint kinase phosphorylation in HCT116 cells exposed to hydroxyurea for 6 h (Fig. 5a).

As PP2A dephosphorylates checkpoint kinases[21–25], we considered that MS-275 modulates PP2A. To test this hypothesis, we used the PP2A inhibitors okadaic acid and cantharidin[51]. While okadaic acid reduced the inhibition of hydroxyurea-induced ATM phosphorylation by MS-275 strongly, it barely prevented the concurrent dephosphorylation of pCHK1 (Fig. 5b).

Cantharidin, which is less specific for PP2A than okadaic acid[51], rescued pATM and pCHK2 in a dose-dependent manner. However, higher doses of cantharidin rescued CHK1 phosphorylation poorly and changed the migration behaviour of pCHK1 (Supplementary Fig. 3a).

These data suggest that PP2A is involved in the MS-275-induced suppression of checkpoint kinase phosphorylation. To identify putative PP2A subunit(s) that mediate(s) checkpoint kinase dephosphorylation, we performed transcriptome profiling of HCT116 cells. We found that MS-275 increased the expression of the *PPP2R3A* gene, which encodes the PP2A regulatory subunit PR130 (Supplementary Table 1). Quantitative real-time PCR (qPCR) verified that the *PPP2R3A* mRNA was significantly upregulated in cells treated with MS-275 or MS-275/hydroxyurea (Fig. 5c).

The time-delayed accumulation of PR130 (Fig. 5c) correlated with the time-delayed inhibition of checkpoint kinase phosphorylation (Fig. 1c). Moreover, PR130 accumulated in cells exposed to MS-275 in a dose-dependent manner, which correlated with reduced checkpoint kinase phosphorylation (Fig. 5d).

HDAC1–3 are the specific targets of MS-275[44]. Hence, a genetic elimination of these HDACs should induce PR130. The simultaneous knockdown of HDAC1 and HDAC2 increased the expression of PR130 (Fig. 5e). As seen with MS-275, hydroxyurea augmented this effect (Figs. 5c–e). Additional elimination of HDAC3 had no further effect on PR130 in hydroxyurea-treated cells (Fig. 5e; Supplementary Fig. 3b, c).

HDAC1 and HDAC2 control the expression of PR130 at the transcriptional level (Figs. 5c). To test whether HDAC1 and HDAC2 are recruited to the *PPP2R3A* promoter, we carried out chromatin immunoprecipitation (ChIP) experiments. We could detect both HDACs at the *PPP2R3A* promoter, at which MS-275 triggered an increase in acetylated histone H3 and a reduction of HDAC1 (Fig. 5f).

In order to examine the relevance of PR130 that is induced upon the knockdown of HDAC1/HDAC2 for checkpoint kinase phosphorylation, we combined RNAi targeting HDACs with hydroxyurea. Attenuating HDAC1 plus HDAC2 decreased the hydroxyurea-induced phosphorylation of ATM and CHK1 (Fig. 5g; Supplementary Fig. 3d). Additional elimination of HDAC3 produced no further effects (Figs. 5e, g).

Next, we expressed PR130 ectopically in HCT116 cells and treated them with hydroxyurea. Overexpression of PR130 reduced ATM phosphorylation similarly to the treatment with MS-275. In contrast, overexpressed PR130 did not attenuate the hydroxyurea-induced phosphorylation of CHK1 (Fig. 5h).

Hence, HDAC1 and HDAC2 are required to repress the expression of PR130 and to maintain checkpoint kinase phosphorylation.

---

**Fig. 2** Loss of checkpoint control in the presence of MS-275. **a** HCT116 cells were treated with 2 μM MS-275 and/or 1 mM hydroxyurea (HU) for 24 h. Cell cycle analysis shown as mean ± SD ($n = 3$). Control cells have G1 54%, S 22%, G2 24%; HU leads to G1 27%, S 55%, G2 18%; MS-275 leads to G1 69%, S 9%, G2 22%; HU/MS-275 leads to G1 30%, S 44%, G2 26%. Statistical significance is displayed for G2 cells (one-way ANOVA, *$P < 0.05$). **b** Treatment with HU (1 mM) and/or MS-275 (2 μM) was carried out as indicated (16–28 h). Representative histograms of cell cycle analysis are shown ($n = 3$). **c** HCT116 cells were treated as in **a**. Immunofluorescence was performed using Tubulin antibody followed by Alexa Fluor-488-coupled secondary antibody; TO-PRO3, nuclear staining. Top: representative pictures of treated cells; scale bar, 20 μm. Bottom: quantitative analysis of mitotic cells using Tubulin immunofluorescence. Data are per cent of mitotic cells relative to untreated cells (mean ± SD; $n = 2$; Student's $t$-test; **$P < 0.01$). **d** DNA fibre assay: CldU (magenta tracks) and IdU (green tracks) were detected using specific primary and secondary antibodies. Replication tracks were classified according to schematic shown (left panel). Representative replication tracks are shown (right panel). **e** Quantitative analysis of replication tracks. Length was calculated in kb pairs/min (2.59 kb pairs = 1 μm). Data are mean ± SD ($n = 3$). **f** Cells were treated with 1 mM HU and 2 μM MS-275 for 6–24 h. Whole-cell extracts were blotted for WEE1, pCDK1 and HSP90 (loading control; $n = 4$). **g** Cells were treated as stated in **f**, immunoblot was done for Cyclin B and mSIN3A (loading control; $n = 3$). **h** Cells were treated as stated in **f**; immunoblot was performed for phosphorylated p53 (S15), p53, WIP1, p21 and Tubulin (loading control; $n = 3$). **i** Amounts of p53 and phosphorylated p53, with their levels in HU-treated cells set as 1. Data were normalised to loading controls. Results represent the mean ± SD ($n = 4$; one-way ANOVA; ***$P < 0.001$)

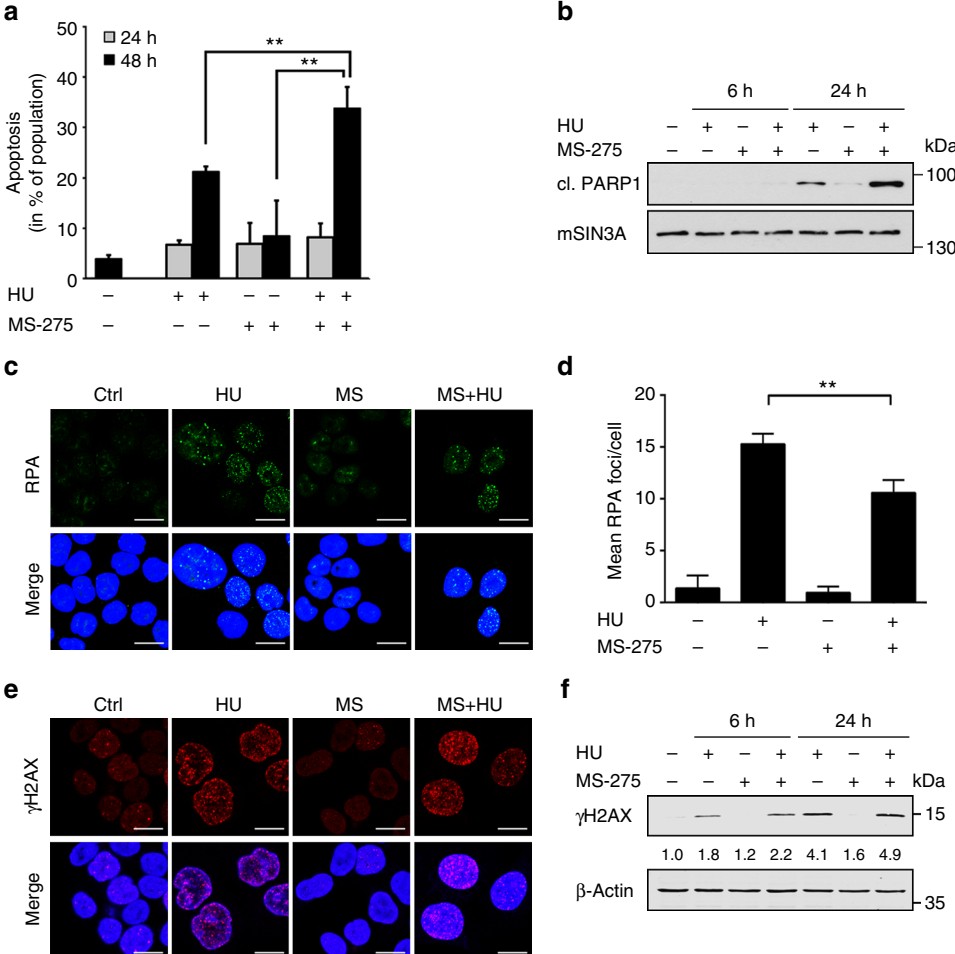

**Fig. 3** DNA damage and apoptosis in the presence of hydroxyurea and MS-275. **a** Flow cytometry analysis of HCT116 cells using Annexin-V-FITC staining. Cells were treated with 1 mM hydroxyurea (HU), 2 µM MS-275 or both for 24–48 h. Results represent the mean ± SD ($n = 3$; one-way ANOVA, **$P <$ 0.01). **b** HCT116 cells were treated with 1 mM HU and 2 µM MS-275 for 6 and 24 h. PARP1 cleavage (cl. PARP1) was analysed by western blot. mSIN3A served as loading control ($n = 3$). **c** HCT116 cells were treated with 1 mM HU and 2 µM MS-275 for 24 h, fixed and incubated with RPA-specific antibody (green). Staining with secondary antibody was performed using Alexa Fluor-488-coupled antibody and TO-PRO3 was used to visualise nuclei. Representative images are shown ($n = 3$; scale bar, 10 µm). **d** Analysis of RPA foci per cell in HCT116 cells. Number of foci was determined using ImageJ software. Data represent the mean ± SD ($n = 3$; one-way ANOVA; **$P < 0.01$). **e** HCT116 cells were cultured with HU (1 mM) and/or MS-275 (2 µM) for 24 h. In the following, samples were fixed, incubated with ɣH2AX antibody (phospho-S139-H2AX) and stained with Alexa Fluor-488-coupled secondary antibody (red). TO-PRO3 was used for nuclear staining. Representative images are shown ($n = 3$; scale bar, 10 µm). **f** Cells were treated with 1 mM HU and 2 µM MS-275 for 6 and 24 h. Whole-cell lysates were analysed for ɣH2AX by western blot. β-Actin was used as loading control. Numbers indicate densitometric analysis of ɣH2AX signals relative to untreated control and normalised to loading control ($n = 3$)

**PR130 controls the phosphorylation of ATM**. To precisely analyse the relevance of PR130 for checkpoint kinase signalling, we eliminated PR130 using the CRISPR-Cas9 technology (Fig. 6a). We obtained several clones devoid of PR130 (HCT116$^{\Delta PR130}$) and PR130-positive control clones (HCT116$^{\Delta gDNA}$) carrying the empty vector (Supplementary Fig. 4a). We analysed two HCT116$^{\Delta PR130}$ clones (#3 and #16), HCT116$^{\Delta gDNA}$ cells and naive HCT116 cells in further analyses.

We tested putative effects of the elimination of PR130 on HDACs and PP2A. Elimination of PR130 had no effect on HDAC1−3 and the equal hyperacetylation of histone H3 in response to MS-275 illustrated that the clones reacted equally to MS-275. Moreover, the elimination of PR130 did not affect the levels of PP2A-A and PP2A-C (Supplementary Fig. 4b, c)

In addition to PR130, MS-275 augmented the mRNA levels of PPP2R5B/B56β; other PP2A-B subunits were not increased (Supplementary Table 1). Therefore, we analysed the expression of this factor and PR48/PPP2R3B as control. MS-275 and

hydroxyurea as well as the elimination of PR130 did not modulate B56β and PR48 significantly (Supplementary Fig. 4c). As expected (Figs. 5c, d), MS-275 induced PR130 in HCT116$^{\Delta gDNA}$ cells (Supplementary Fig. 4d). These results validated our newly generated cellular models.

Next, we treated them with hydroxyurea and MS-275 and analysed checkpoint kinase phosphorylation. Because of the relevance of ATM and CHK1 for cell survival during replicative stress (Fig. 4) and their dephosphorylation in the presence of MS-275 (Figs. 1a–d), we focused on these kinases. MS-275 decreased the hydroxyurea-induced phosphorylation of ATM in HCT116$^{\Delta gDNA}$ cells. In HCT116$^{\Delta PR130}$ cells, MS-275 did not significantly affect the phosphorylation of ATM (Fig. 6b). Furthermore, when we decreased PR130 with short interfering RNAs (siRNAs), we could corroborate that PR130 is required for ATM dephosphorylation in the presence of hydroxyurea and MS-275 (Fig. 6c).

Next, we probed for an interaction between PR130 and pATM in cells treated with hydroxyurea and MS-275. To

prevent dephosphorylation of ATM under these conditions (Figs. 1a–c and 5b), we applied okadaic acid. We could co-precipitate pATM in complexes with PR130 (Fig. 6d). When we incubated HCT116 cells with okadaic acid alone or with MS-275/hydroxyurea, we found that MS-275 promoted the interaction between immunoprecipitated pATM and PR130 (Fig. 6d).

Since MS-275 modulates protein acetylation, we tested for an acetylation of PR130 in the presence of MS-275. We detected that MS-275 caused a 4.2-fold increase of the basal acetylation of PR130 (Fig. 6e).

To verify that PR130 from HCT116 cells treated with MS-275 is catalytically active, we expressed HA-PR130 in such cells, immunoprecipitated it and tested its activity against a

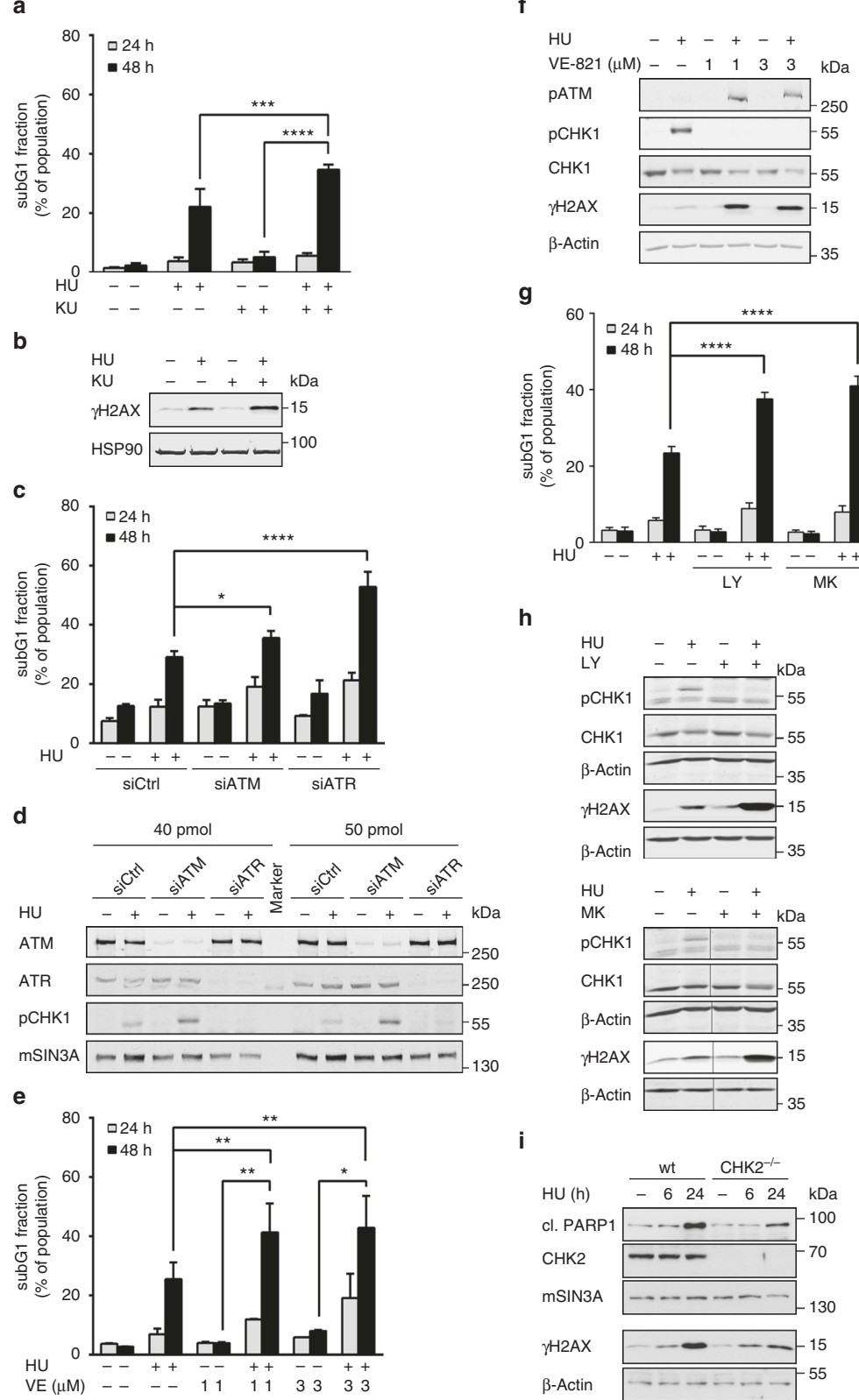

phosphorylated peptide surrounding S1981 of ATM. Immuno-precipitated HA-PR130 complexes catalysed the dephosphorylation of this peptide (Fig. 6f). Accordingly, PR130 from cells treated with MS-275 binds PP2A-A and -C subunits (Supplementary Fig. 4e). Despite reported constitutive interactions between ATM and PP2A-A and -C subunits in human lymphoblastoid cells[24], we could not detect such an interaction in HCT116 colon cancer cells and K562 leukaemia cells (Supplementary Fig. 4f).

These data suggest that PR130 promotes the PP2A-dependent dephosphorylation of ATM in cells with replicative stress.

**PR130 dictates replicative arrest and CHK1 phosphorylation**. Next, we analysed pCHK1 in HCT116$^{\Delta gDNA}$ and HCT116$^{\Delta PR130}$ cells that we exposed to hydroxyurea and MS-275. We found that HCT116$^{\Delta PR130}$ cells had significantly higher levels of pCHK1 and that MS-275 attenuated the phosphorylation of CHK1, but not of ATR, in both HCT116$^{\Delta gDNA}$ and HCT116$^{\Delta PR130}$ cells (Figs. 7a,b).

These data suggest that PR130 antagonises the phosphorylation of ATM and CHK1 via two independent molecular mechanisms. Because of the regulation of CHK1 phosphorylation by cell cycle regulatory molecules[7,18,19,52], we analysed cell cycle control in response to hydroxyurea and MS-275 in HCT116$^{\Delta gDNA}$ and HCT116$^{\Delta PR130}$ cells. Upon replicative stress, the number of HCT116$^{\Delta gDNA}$ cells in S phase increased from 18 to 43%, while the percentage of HCT116$^{\Delta PR130}$ cells in S phase increased from 17 to 36% (Fig. 7c). Addition of MS-275 to hydroxyurea allowed the progression of 7% of HCT116$^{\Delta gDNA}$ cells and of 6% HCT116$^{\Delta PR130}$ cells from S to G2 phase (Fig. 7c).

We then compared the WEE1-dependent phosphorylation of CDK1 and the levels of p21 in HCT116$^{\Delta PR130}$ and HCT116$^{\Delta gDNA}$ cells. We noted that in response to hydroxyurea, HCT116$^{\Delta PR130}$ cells had lower levels of pCDK1 (Fig. 7d) and higher levels of p21 (Fig. 7e). This finding suggests that an earlier G1 arrest of hydroxyurea-treated HCT116$^{\Delta PR130}$ cells is due to an augmented phosphorylation of p53, which activates p21 expression (Figs. 7c and 6b). Upon replicative stress, MS-275 attenuated WEE1/pCDK1 signalling and p21 in both cell types (Figs. 7d, e) and this can explain their transitions from S to G2/M phase (Fig. 7c). Phosphorylation of HDAC2 was not changed significantly in cells with or without PR130 (Fig. 7d).

These data let us hypothesise that an earlier arrest in S phase can explain the higher phosphorylation of CHK1 in cells lacking PR130. To test this idea, we applied increasing doses of hydroxyurea and analysed cell cycle arrest, p21 levels, pCHK1 and WEE1/pCDK1. An earlier arrest in S phase together with an increased accumulation of p21 occurred when HCT116 cells were treated with increasing doses of hydroxyurea; levels of pCDK1

were not augmented further. Consistent with our hypothesis, this led to a dose-dependent increase of pCHK1 (Figs. 7f, g).

To test our hypothesis of a cell cycle-dependent control of pCHK1 further, we analysed checkpoint kinase signalling in response to γ-irradiation (γ-IRR). In this setting, the WEE1/pCDK1 checkpoint stalls cell cycle progression[7,18]. If the suppression of WEE1/pCDK1 by MS-275 (Figs. 2f and 7d) is associated with a reduced phosphorylation of CHK1, MS-275 should also impair WEE1 and the phosphorylation of CHK1 in HCT116 cells exposed to γ-IRR. Indeed, MS-275 reduced the γ-IRR-induced phosphorylation of CHK1 and WEE1 levels. MS-275 also reduced the induction of pCHK2 and pp53, but the γ-IRR-induced, auto-phosphorylated ATM was insensitive to MS-275 (Supplementary Fig. 4g, h).

These findings position PR130 as upstream regulator of pCHK1, WEE1/pCDK1 and p21, and consequently as a new modulator of cell cycle progression during replicative stress.

**PR130 affects DNA damage and controls apoptosis**. The analysis of RPA and γH2AX foci in HCT116$^{\Delta gDNA}$ and HCT116$^{\Delta PR130}$ cells revealed that, despite their slower cell cycle progression upon replicative stress (Fig. 7c), HCT116$^{\Delta PR130}$ cells carried higher numbers of RPA foci (28 vs. 17 foci/cell) and γH2AX foci (1.5-fold vs. 2.3-fold increase), indicating more pronounced replicative stress (Figs. 8a–c).

Treatment with MS-275 decreased RPA foci in hydroxyurea-treated HCT116$^{\Delta gDNA}$ and HCT116$^{\Delta PR130}$ cells to 13 and 20 foci/cell, respectively (Figs. 8a, b). This decrease of RPA foci was linked to persistent levels of γH2AX and a reduction of RAD51 (Figs. 8c, d; Supplementary Fig. 5a).

These data illustrate that, while MS-275 reduces RPA foci indicating ssDNA, γH2AX foci indicating damaged DNA persist. Furthermore, compared to PR130-positive cells, a larger portion of HCT116$^{\Delta PR130}$ cells underwent apoptosis when exposed to hydroxyurea and MS-275 (Figs. 8e, f; Supplementary Fig. 5b).

Taken together, our results demonstrate that a loss of PR130 leads to an early arrest in S phase, increased replicative stress and augmented sensitivity to MS-275.

**Discussion**
Our report reveals that the epigenetic modifiers HDAC1 and HDAC2 control checkpoint kinase phosphorylation through a suppression of PR130. HDACi as well as an elimination of HDAC1/HDAC2 induce PR130-dependent mechanisms that restrict checkpoint kinase phosphorylation (Supplementary Fig. 6).

A sustained phosphorylation of ATM and CHK1 during replicative stress requires HDAC1 and HDAC2. These bind to the

**Fig. 4** Checkpoint kinases ensure cell survival upon replicative stress. **a** HCT116 cells were pretreated with 3 μM KU-60019 for 1 h followed by stimulation with 1 mM hydroxyurea (HU) for 24–48 h. Cell death was detected using PI staining and flow cytometry. Results are displayed as mean ± SD ($n_{24 h} = 3$, $n_{48 h} = 4$; one-way ANOVA, ***$P < 0.001$; ****$P < 0.0001$). **b** Western blot analysis of HCT116 cells treated with KU-60019 and HU for 24 h as described in **a**. γH2AX was detected with specific antibodies. HSP90 served as loading control. **c** siRNA transfection was performed as described in the Methods section. Twenty-four hours after transfection, HCT116 cells were incubated with 1 mM HU for 24–48 h. Cell death analysis was performed as described in **a**. Data are presented as mean ± SD ($n = 4$; one-way ANOVA, *$P < 0.05$; ****$P < 0.0001$). **d** Transfection with indicated siRNAs and treatment were performed as described in **c**. ATM, ATR, pCHK1 (S317) and mSIN3A (loading control) were analysed by immunoblotting. **e** HCT116 cells were treated with 1 mM HU for 24–48 h after 1 h pre-incubation with 1–3 μM VE-821. SubG1 fractions were measured as described in **a**. Results represent mean ± SD ($n = 2$; one-way ANOVA, *$P < 0.05$, **$P < 0.01$). **f** HCT116 cells were treated with VE-821 and/or HU for 24 h as described in **e**. Western blot analysis was performed to detect pATM (S1981), pCHK1 (S317), CHK1 and γH2AX. β-Actin served as loading control. **g** Cells were pre-incubated with 300 nM LY2603618 or MK-8776 for 1 h followed by HU treatment (1 mM) for 24–48 h. Cell death was analysed as described in **a**. Results are displayed as mean ± SD ($n_{24 h} = 4$, $n_{48 h} = 3$; one-way ANOVA, ****$P < 0.0001$). **h** Western blot analysis of HCT116 treated for 24 h as described in **g**. γH2AX, pCHK1 (S296) and CHK1 levels were detected with specific antibodies. β-Actin served as loading control. **i** HCT116 cells and their comparative CHK2$^{-/-}$ cell line were treated with 1 mM HU for the indicated time periods. Indicated proteins were detected via western blot

*PR130* promoter and MS-275 increases histone acetylation at this locus. MS-275 also reduces the amount of HDAC1 at the *PR130* promoter. As HDACs do not bind DNA directly, HDAC1 may bind to a transcription factor that detaches from DNA in the presence of MS-275. Irrespective of any remaining binding of HDAC1/HDAC2, MS-275 can increase the acetylation of the *PR130* promoter together with the expression of PR130.

Experiments in which we genetically eliminated PR130 verify that this PP2A-B subunit is necessary for the dephosphorylation of ATM when HDAC1 and HDAC2 are inhibited in cells with replicative stress. The induction of PR130 by MS-275 is accompanied by the formation of a complex between pATM and PR130

and an increase of acetylated PR130. Immunoprecipitated PR130 from cells treated with hydroxyurea plus MS-275 contains PP2A-A/PP2A-C. Accordingly, such PR130 complexes have catalytic activity against a phosphorylated ATM peptide and okadaic acid prevents ATM dephosphorylation in hydroxyurea/MS-275-treated cells. Our finding that hydroxyurea does not trigger an interaction between pATM and PR130 demonstrates that ATM phosphorylation is necessary but not sufficient for its association with PR130. This observation corresponds well with the possibility to induce pATM with hydroxyurea and with about equal levels of pATM in hydroxyurea-treated HCT116$^{\Delta PR130}$ and HCT116$^{\Delta gDNA}$ cells. Apparently, MS-275 allows the interaction

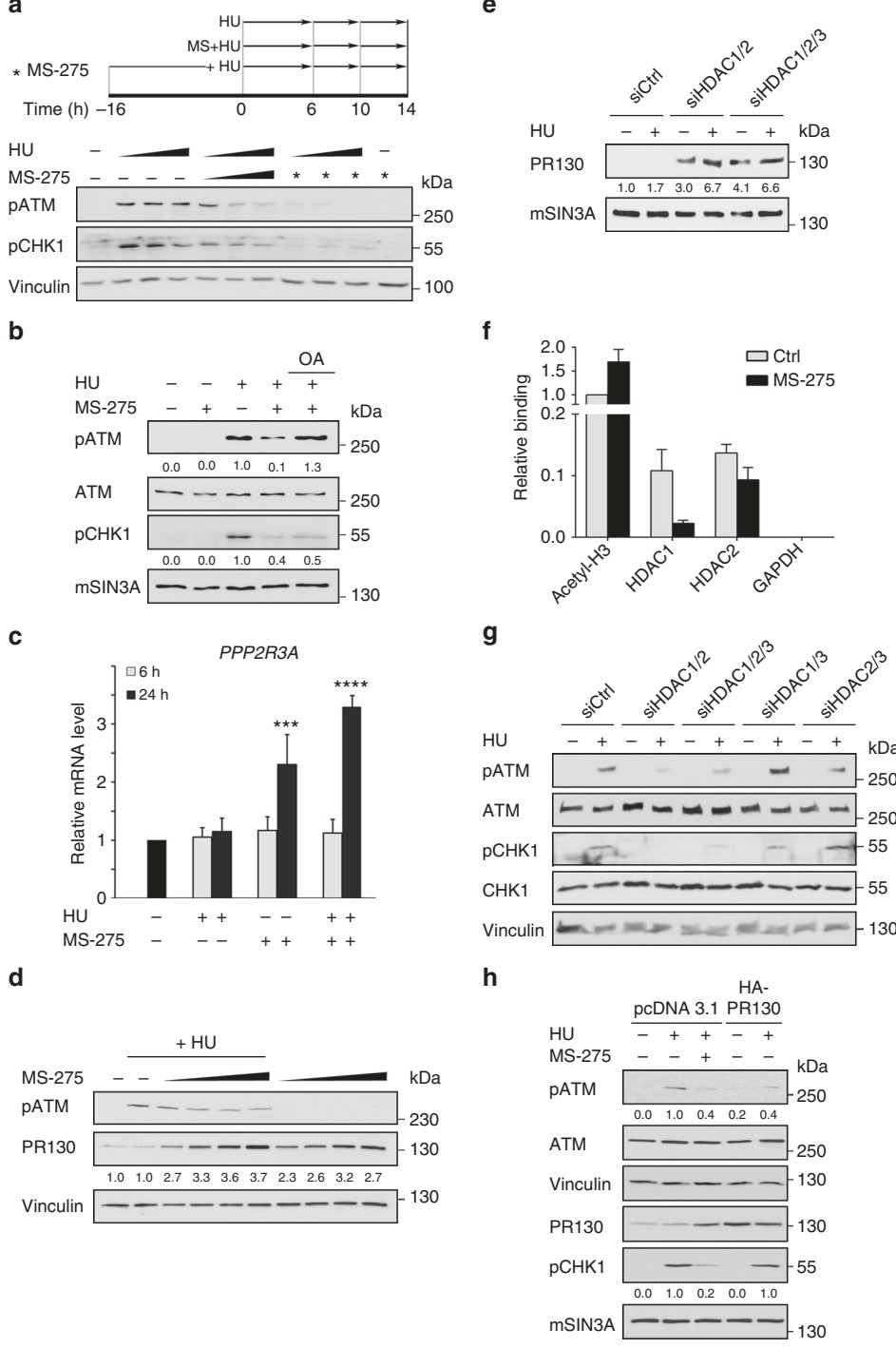

between pATM and PR130 for ATM dephosphorylation by the catalytic PP2A-C subunit. MS-275 might enhance ATM dephosphorylation through a hyperacetylation of PR130, but the levels of PR130 in untreated cells may likewise be insufficient to mediate ATM dephosphorylation. Further studies will answer these questions.

We also considered the possibility that hydroxyurea disrupts a reported basal interaction between PP2A-A and PP2A-C with ATM[24]. However, we could not detect such an interaction in HCT116 cells and in K562 cells. These data are consistent with literature reports that suggest no constitutive inactivation of ATM by PP2A[23,25,26]. Nonetheless, we do not want to exclude that other PP2A-B subunits regulate the dephosphorylation of ATM in other settings. For example, the B55 subunit of PP2A promotes the dephosphorylation of ATM in cells infected with Adenovirus[53].

A reduced sensitivity of ATR to MS-275 can explain why hydroxyurea triggers ATM phosphorylation in MS-275-treated HCT116$^{\Delta PR130}$ cells. In such cells there is no PR130-PP2A that could antagonise the hydroxyurea-evoked, ATR-dependent phosphorylation of ATM. In contrast to hydroxyurea, γ-IRR induces ATM autophosphorylation at S1981 that is insensitive to MS-275. Perhaps, the interaction of ATM with the MRN complex at DSBs[1,2,7] prevents access of PR130-PP2A to ATM. Congruent with this idea, the MRN complex is not involved in the ATR-mediated phosphorylation of ATM[10,48].

The HDACi-mediated attenuation of CHK1 phosphorylation in hydroxyurea-treated HCT116$^{\Delta PR130}$ and HCT116$^{\Delta gDNA}$ cells seems not to rely on a direct dephosphorylation of CHK1 by PR130-PP2A. The higher levels of pCHK1 in hydroxyurea-treated HCT116$^{\Delta PR130}$ cells compared to HCT116$^{\Delta gDNA}$ cells rather appear to result from an earlier S phase arrest of HCT116$^{\Delta PR130}$ cells. In agreement with this hypothesis, MS-275 promotes cell cycle progression and reduces this hyperphosphorylation of CHK1 independent of PR130. Accordingly, overexpressed PR130 attenuates the hydroxyurea-evoked phosphorylation of ATM, but not of CHK1. Further evidence for a dominant role of cell cycle progression for CHK1 phosphorylation can be deduced from higher levels of pCHK1 and p21 that accumulate when increasing doses of hydroxyurea are applied to HCT116 cells. High levels of pCHK1, pp53 and p21 can likewise explain the slow G1−S phase transition of HCT116$^{\Delta PR130}$ cells treated with hydroxyurea. Such data correspond well with the literature[7,52]. Moreover, reduced pCHK1 in hydroxyurea-treated cells with a knockdown of HDAC1/HDAC2 agrees with a p21-dependent cell cycle arrest that is seen in cells lacking HDAC1/HDAC2 (refs. [54,55]). The reduction of WEE1/pCDK1 by MS-275 could also attenuate the hydroxyurea-induced CHK1 phosphorylation and thereby pp53.

This idea is supported by the finding that WEE1 sustains CHK1 activation during gemcitabine-induced replicative stress[19]. Other regulatory molecules may also control the phosphorylation of CHK1 and, of course, our data do not contradict a dephosphorylation of CHK1 by PP2A in other settings. It will be interesting to identify the targets of the PR130–PP2A complexes that modulate cell cycle progression upon replicative stress.

Although hydroxyurea-treated HCT116$^{\Delta PR130}$ cells arrest earlier at the G1/S phase boundary, they lose cell cycle control in response to MS-275. This process is linked to a modulation of pCHK1, WEE1/pCDK1, pp53/p53 and p21. While hydroxyurea and MS-275 induce p21, their combination attenuates p21. The induction of p21 by MS-275 and the following G1 arrest are well in line with the known repression of p21 by HDAC1/HDAC2 (refs. [54,55]). The accumulation of p21 in hydroxyurea-treated HCT116 cells is consistent with the literature[56] and is linked to p53. Reduced levels of p21 in the hydroxyurea/MS-275 co-treatment can be explained by a reduced activity of p53 and there is a proteasome- and caspase-dependent attenuation of HDACi-induced p21 in cells that are co-treated with HDACi and hydroxyurea[45]. This decrease in p21 is associated with increased Cyclin B levels in hydroxyurea/MS-275-treated cells. As p21 inhibits CDK1/Cyclin B complexes, these data together with the inhibitory effect of MS-275 on the cell cycle restricting WEE1/pCDK1 node can explain why such cells traverse from S to G2/M phase.

Our data further demonstrate that ATM activation is not sufficient to maintain S phase arrest in the presence of hydroxyurea and MS-275. Nevertheless, ATM, ATR and CHK1 ensure cell survival and prevent DNA damage upon replicative stress. ATR prevents a global exhaustion of the RPA pool and thereby averts the formation of γH2AX-positive DSBs[12]. We further reveal that the ATR targets CHK1 and ATM[10] ensure cell survival. The ATM target CHK2 seems to have an opposite role, which agrees with previous findings showing that CHK2 and ATM have overlapping as well as divergent functions[57].

Compared to HCT116$^{\Delta gDNA}$ cells, lower numbers of HCT116$^{\Delta PR130}$ cells enter S phase with hydroxyurea. Nevertheless, these lower amounts of PR130-negative cells carry even more replicative stress-associated RPA foci and γH2AX than HCT116$^{\Delta gDNA}$ cells. These higher levels of replicative stress could stem from a reduced inactivation of CDK1 by WEE1, as CDK1 can promote unscheduled origin firing[7,18]. Nonetheless, HCT116$^{\Delta PR130}$ cells undergo apoptosis to the same extent as HCT116$^{\Delta gDNA}$ cells when treated with hydroxyurea. Hyperphosphorylation of CHK1 in HCT116$^{\Delta PR130}$ cells, enhanced activation of the intra S phase checkpoint and HR likely prevent apoptosis. Addition of MS-275 to hydroxyurea-treated

**Fig. 5** PR130 upregulation by MS-275 represses checkpoint kinase signalling. **a** HCT116 cells were stimulated with 1 mM hydroxyurea (HU) and/or 2 μM MS-275. Wedges signify increasing incubation times (6, 10 and 14 h). Asterisks signify pre-incubation with MS-275 for 1 h. Indicated (phosphorylated) proteins were analysed by western blot. Vinculin served as loading control. **b** HCT116 cells were cultured with 2 μM MS-275 and/or 1 mM HU for 24 h and incubated with the PP2A inhibitor ocadaic acid (25 nM) for additional 4 h. Phosphorylated forms of ATM, CHK1 and mSIN3A (loading control) were detected by immunoblot. Numbers indicate densitometric analysis of western blot signals relative to HU-treated samples and normalised to loading control. **c** Relative mRNA levels of *PPP2R3A* in HCT116 cells evaluated by quantitative RT-PCR. Cells were treated with 1 mM HU and/or 2 μM MS-275 (6–24 h). Results show the mean ± SD (n = 3; one-way ANOVA, ***P < 0.001; ****P < 0.0001). **d** Cells were cultured with or without increasing amounts of MS-275 (0.5; 1; 2; 5 μM) alone or in combination with 1 mM HU for 24 h. Indicated proteins were detected by immunoblot. Densitometric analysis of PR130 signals is indicated relative to untreated cells and normalised to Vinculin levels (n = 3). **e** Knockdown HDAC1–3 was performed with siRNAs by transient transfection for 48 h. 1 mM HU was added for further 24 h. PR130 was detected by immunoblotting, quantified relative to untreated siCtrl cells and normalised to mSIN3A levels (loading control). **f** ChIP was performed to detect acetylated histone H3 and the binding of HDAC1/HDAC2 to the *PPP2R3A* promoter in presence/absence of MS-275. Acetylation of histone H3 is set as 1. Results show the mean ± SD (n = 3). **g** Cells were transiently transfected and treated as described in **e**. Indicated (phosphorylated) proteins were detected by western blot. **h** HCT116 cells were transfected with pcDNA3.1 or HA-PR130 for 48 h and treated with 1 mM HU and 2 μM MS-275 for additional 24 h. Indicated proteins were detected by immunoblotting. Western blot signals were quantified relative to HU-treated cells and normalised to loading controls (Tubulin, Vinculin; n = 2)

HCT116$^{\Delta PR130}$ cells attenuates CHK1 phosphorylation and leads to ongoing DNA synthesis and lethal cell cycle progression into mitotic catastrophe and apoptosis. Since there is non-oncogene addiction to high CHK1 activity[58], the hyperphosphorylation of CHK1 in HCT116$^{\Delta PR130}$ cells may

sensitise them to MS-275. We further show that MS-275 attenuates RAD51, an essential HR protein[16,43]. Depletion of RAD51 is probably more detrimental in HCT116$^{\Delta PR130}$ cells than in HCT116$^{\Delta gDNA}$ cells, because HCT116$^{\Delta PR130}$ cells are subject to increased replicative stress. Congruently, the reduction of

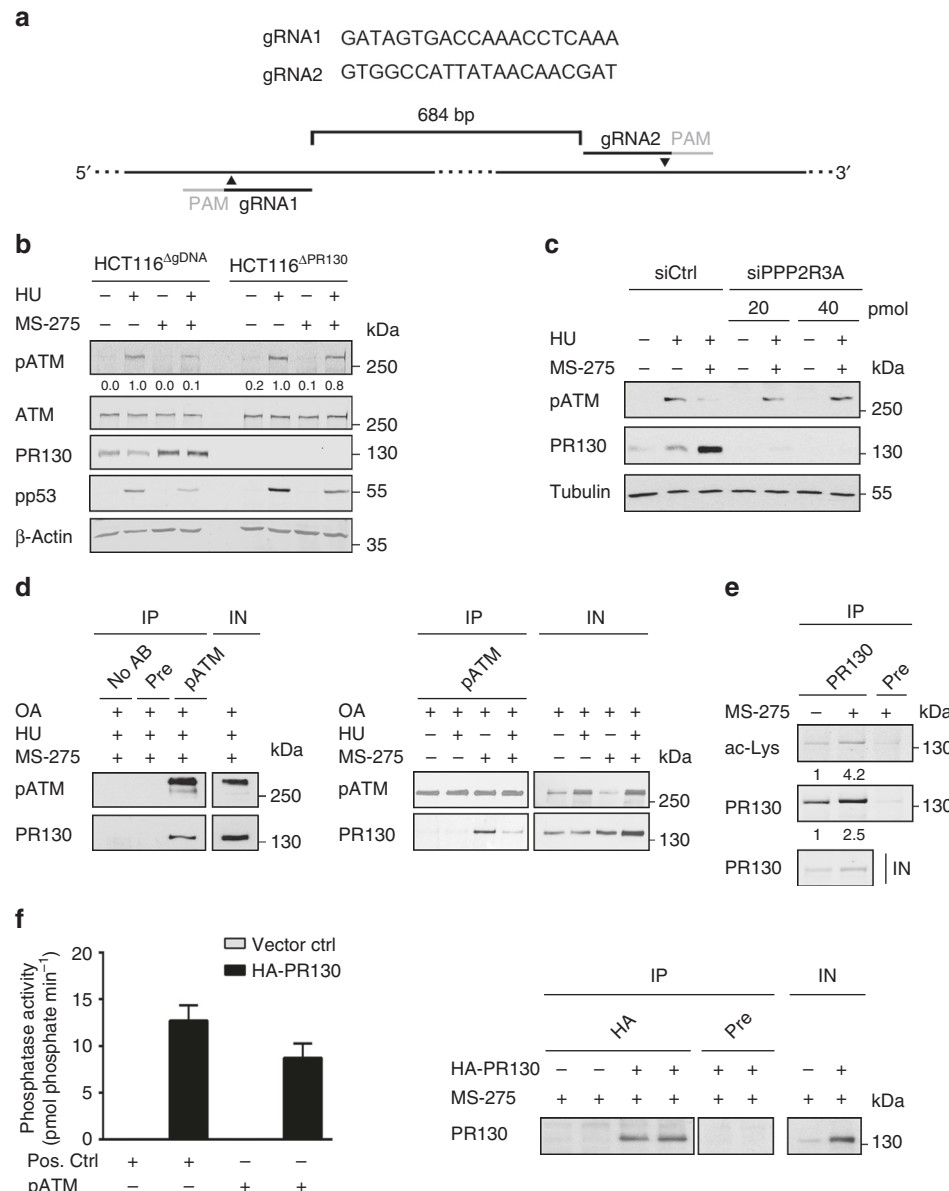

**Fig. 6** PR130 expression antagonises ATM phosphorylation. **a** Schematic of *PPP2R3A* knockout using CRISPR-Cas9 technology. Humanised *S. pyogenes* Cas9 nuclease was expressed in HCT116 cells with two guide RNAs (gRNA) to introduce two DSB into the *PPP2R3A* gene (684 bp apart) as described in the Methods section. The usage of two cleavage sites increased the chances to create a successful knockout. **b** HCT116$^{\Delta gRNA}$ and HCT116$^{\Delta PR130}$ (clone #16) cells were cultured with 2 µM MS-275 and/or 1 mM hydroxyurea (HU) for 24 h. ATM, pATM (S1981), p53, pp53 (S15) and PR130 were detected by western blot. β-Actin served as loading control. Numbers indicate densitometric analysis of pATM signal relative to HU-treated sample of each cell line and normalised to β-Actin signal (n = 3). **c** HCT116 cells were transiently transfected with indicated siRNAs. After 24 h, cells were treated with 1 mM HU and 2 µM MS-275 (24 h). Protein detection was performed by immunoblot (n = 2). **d** After 20 h of treatment with HU (1 mM) and MS-275 (2 µM), okadaic acid (OA, 25 nM) was added for further 4 h. Co-immunoprecipitations (IP) with anti-pS1981-ATM antibody, rabbit pre-immune serum (pre) or no antibody (no AB) were analysed for pATM and PR130 by western blot. Input (IN) represents 6% of the lysates used for IP. The right panel shows results that were obtained with the same strategy, including individual treatment with HU and MS-275. Data represent results from three independent experiments that were carried out in a similar manner. **e** IP performed with PR130 antibody using lysates from untreated and MS-275-treated HCT116. Precipitates were probed with pan-acetyl-lysine (ac-Lys) and PR130 antibody. Input (IN) is 6% of the lysate used for immunoprecipitation (n = 2). Numbers indicate densitometric analysis of ac-Lys and PR130 signal relative to untreated cells. **f** HCT116 cells were transfected with HA-PR130 or vector control (pcDNA3.1) for 24 h and subsequently treated with MS-275 (2 µM, 24 h). Phosphatase activity of the HA-precipitates against phospho-S1981-ATM peptide or a threonine phosphopeptide (positive control) were measured in vitro as liberated pmol phosphate/min. Data are presented as mean ± SEM (n = 3). The western blot verifies precipitation of HA-tagged PR130

RAD51 by MS-275 hampers DNA repair, leading to persistent γH2AX foci and apoptosis. Truly, MS-275 may trigger further cell death pathways, such as p53/NF-κB signalling[43].

While our results illustrate a critical role of class I HDACs and PR130 for S phase arrest, other mechanisms might be independent or could crosstalk with PP2A/PR130. An example might be the BRCA2-dependent regulation of RAD51[59]. WIP1 also inactivates CHK1 and ATM[60,61], but MS-275 decreases WIP1 protein levels. Our finding that cantharidin but not okadaic acid can rescue pCHK1 in cells treated with hydroxyurea plus MS-275 may also hint to an involvement of other phosphatases[51]. Although previous studies found that HDACi reduces CHK1 and ATM[34,62,63], our data demonstrate no significant loss of these proteins in response to HDACi. Hence, in our cancer-derived systems, the chemical or genetic inactivation of HDACs unlikely impairs checkpoint kinase activation via degradation.

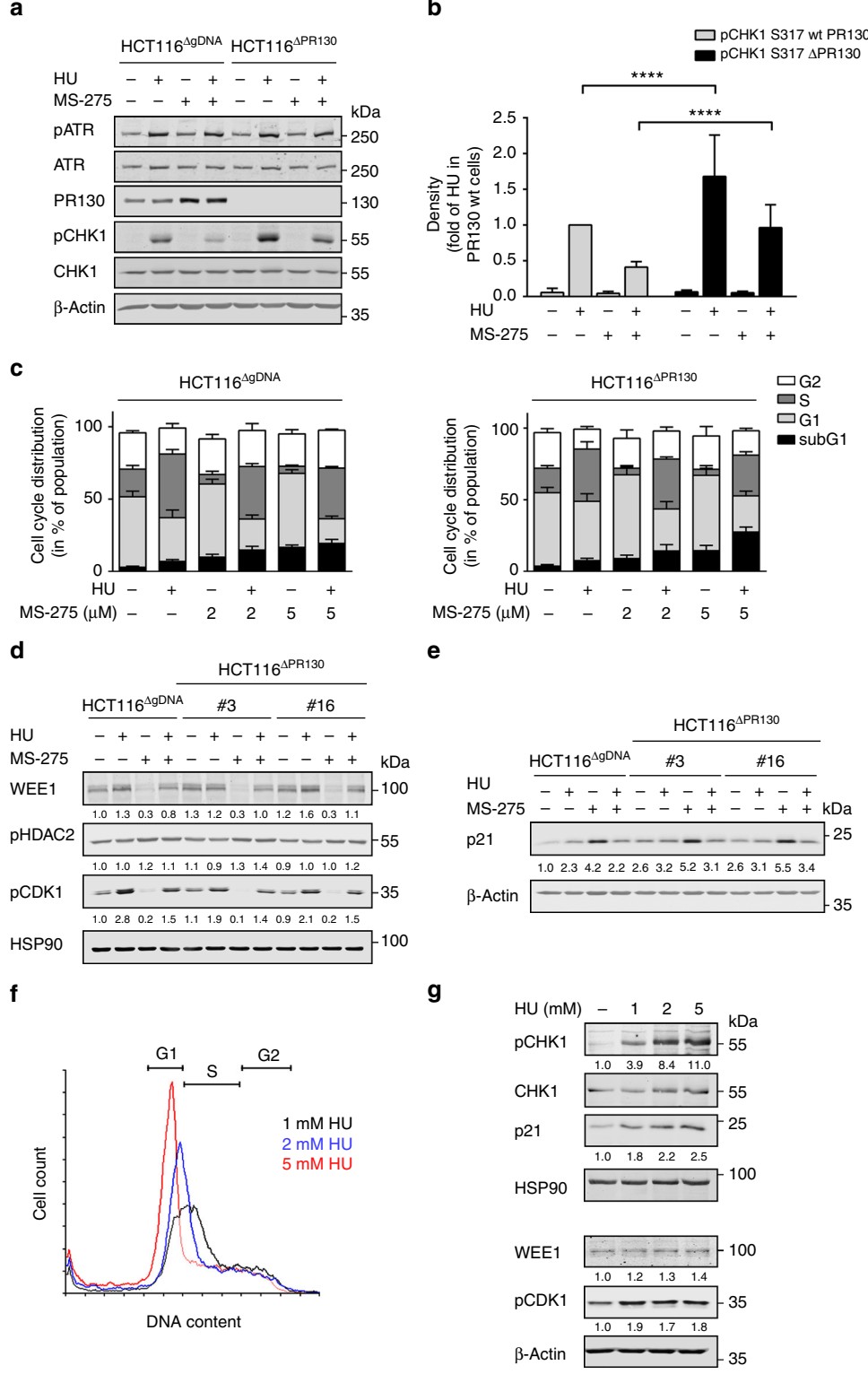

We also show that MS-275 attenuates the N-terminal phosphorylation of p53 by hydroxyurea. While such an attenuation of p53 phosphorylation appears counterintuitive regarding apoptosis induction by hydroxyurea plus MS-275, it appears logical in light of the fact that p53 prevents cell cycle progression upon dNTP shortage and DNA damage[1,7]. Moreover, p53 lacking N-terminal phosphorylation sites retain tumour suppressor activity[64]. It should also be considered that the inhibition of ribonucleotide reductase (RNR) by hydroxyurea is reversible and stalled replication forks restart in the absence of checkpoint signalling[13,16,59,65]. In addition, RNR consists of the large RNR1 (RRM1) and small RNR2 (RRM2 and RRM2B/p53R2) subunits[66] and RRM2 and RRM2B/p53R2 are induced in HU/MS-275-treated HCT116 cells. Moreover, stalled cells can gain dNTPs and other metabolites from the ingestion of surrounding apoptotic cells, which may facilitate S phase progression. Misincorporation of NTPs[67] during checkpoint kinase inactivation by MS-275 could equally allow cell cycle progression into mitotic catastrophe. Consistent with this idea, checkpoint kinases prevent mitotic catastrophe in doxorubicin-treated cancer cells[68], and combinations of the DSB inducer irinotecan and CHK1 inhibitors efficiently eliminate tumour cells[69].

A loss of PR130 promotes the transformation of human cells[30], the *PPP2R3A* gene is often inactivated by DNA methylation[70] and PR130 contributes to cancer-relevant signalling (refs. [29–32]). HDAC1 and HDAC2 are also frequently dysregulated in tumours[42]. Thus, a disturbed expression of PR130 through an aberrant regulation of HDAC1/HDAC2 may affect checkpoint kinase phosphorylation and the fate of cells with compromised genomic integrity[8,9]. Notably, mice with a T-cell-specific deletion of *Hdac1* and one allele of *Hdac2* developed a neoplastic transformation of immature T cells with genomic instability[37,38]. Knocking down *Hdac1/Hdac2* also promotes acute myeloid leukaemia and genomic instability[39]. Tumour-suppressive functions of HDAC1/HDAC2 in the skin became evident when one allele of *Hdac2* was eliminated in *Hdac1* null mice[40]. Our data provide a possible framework for tumour-relevant functions of HDAC1 and HDAC2. On the one hand, inactivation of these HDACs abrogates checkpoint kinase signalling and increases the sensitivity of cancer cells to replicative stress. On the other hand, tumour barrier functions of replicative stress and DNA damage signalling networks might be lost when HDAC1/HDAC2 are absent or inactivated.

## Methods

**Antibodies**. The following antibodies were used during this study: anti-p-Ser1981-ATM (ab81292; 1 : 750), anti-ATM (ab32420; 1 : 1000), anti-p-Thr68-CHK2 (ab32148; 1 : 1000), anti-HDAC1 (ab46985, 1 : 1000), anti-HDAC3 (ab16047; 1 : 1000) from Abcam; anti-p-Thr68-CHK2 (AP01556PU-N; 1 : 500) from Acris Antibodies; anti-Cyclin B1 (ARP30161; 1 : 2000) from Aviva, anti-BrdU (347580; 1 : 1500), anti-cleaved-PARP1 (552596; 1 : 5000) from BD Pharmingen; anti-p-Ser317-CHK1 (A300-163A; 1 : 1000), anti-WIP1 (A300-664A; 1 : 5000) from Bethyl Laboratories; anti-Vinculin (BZL-03106; 1 : 5000) from Biozol; anti-acetyl-Lysine (9681; 1 : 1000), anti-ATR (2790; 1 : 1000), anti-p-Tyr15-cdc2(CDK1) (4539; 1 : 1000), anti-p-Ser317-CHK1 (2344; 1 : 1000), anti-p-Ser296-CHK1 (2349; 1 : 1000), anti-CHK1 (2360; 1 : 1000), anti-p-Thr68-CHK2 (2661; 1 : 500), anti-CHK2 (2662; 1 : 1000), anti-p-Ser139-H2AX (9718; 1 : 1000), anti-p-Ser15-p53 (9284; 1 : 5000), anti-PP2Ac (2259; 1 : 1000) from Cell Signalling Technology; anti-p-Thr1989-ATR (GTX128145; 1 : 1000), anti-PPP2R3B (GTX47038; 1 : 750) from GeneTex; anti-acetyl-Histone3 (06-599; 1 : 2000), anti-ATM (PC116; 0.8 µg), anti-pS421/S423-HDAC2 (07-1575; 1 : 1000), anti-RPA (NA19L; 1 : 1000) from Millipore; anti-ATM (NB100-309; 1 : 1000), anti-PPP2R3A (NBP1-87233; 1 : 2000) from Novus Biologicals; anti-BrdU (OBT0030; 1 : 1000) from Oxford Biotechnologies; anti-p-Ser1981-ATM (#200-301-400; 1 : 1000) from Rockland Immunochemicals; anti-beta-Actin (sc-47778; 1 : 5000), anti-B56β (sc-515676; 1 : 1000), anti-CHK1 (sc-8408; 1 : 2000), anti-CHK2 (sc-9064; 1 : 1000), anti-GAPDH (sc-32233; 1 : 2000), anti-HA (sc-7392; 1 : 1000), anti-HDAC2 (sc-7899; 1 : 2000), anti-HSP70 (sc-24; 1 : 2000), anti-mSIN3A (sc-994; 1 : 20,000), anti-p21 (sc-6246; 1 : 2000), anti-p53 (sc-81168; 1 : 50,000), anti-PP2A-Aα/β (sc-13600; 1 : 1000), anti-PPP2R3A (sc-6115), anti-WEE1 (sc-5285; 1 : 750) from Santa Cruz Biotechnology; anti-Tubulin (T5168; 1 : 50,000), anti-Actin (A2066; 1 : 10,000) from Sigma-Aldrich; anti-acetyl-Histone3 (06–599; 1 : 1000) from Upstate. Secondary antibodies HRP-coupled anti-mouse (7076; 1 : 5000) and anti-rabbit (7074; 1 : 5000) were purchased from Cell Signalling, while IRDye® 680RD- (mouse: 925–68070; 1 : 10,000; rabbit: 925-68071; 1 : 10,000) or IRDye® 800CW-coupled secondary antibodies (mouse: 925-32210, 1 : 10,000; rabbit: 925-32211, 1 : 10,000) were obtained from Licor.

**Drugs and chemicals**. Hydroxyurea, 5-Fluorouracil, cantharidin and propidium iodide were purchased from Sigma-Aldrich; MS-275, KU-60019, MK-8776, LY2603618, LBH589 and VE-821 from Selleck Chemicals; okadaic acid was purchased from Enzo Life Sciences; Annexin-V-FITC-conjugated was from ImmunoTools. Fmoc-protected amino acids and coupling reagents (HBTU, HOBt) were purchased from Novabiochem, Iris Biotech GmbH and Orpegen Pharma GmbH, respectively. Reagents for synthesis and solvents for chromatography (analytical grade) were from VWR International GmbH.

**Cell culture**. HCT116, RKO and MEF cells were cultured in Dulbecco's modified Eagle's medium (GE Healthcare) and K562 cells were cultivated in RPMI 1640 medium (GE Healthcare). Culture media were supplemented with 10% fetal calf serum (Sigma-Aldrich) and 50 µg/ml Gentamicin (GE Healthcare) or 100 U/ml penicillin and 100 µg/ml streptomycin (Thermo Fisher). No cell lines used in this study were found in the database of commonly misidentified cell lines that is maintained by ICLAC and NCBI Biosample. HCT116, RKO and K562 cells were verified by DNA fingerprint at DSMZ, Braunschweig. Cell lines were regularly tested by PCR to exclude mycoplasma contaminations.

**CRISPR-Cas9**. The pX330-U6-Chimeric_BB-CBh-hSpCas9 plasmid was provided by Dr. Feng Zhang (also available on Addgene, plasmid # 42230). Both gRNAs against specific sequences in the *PPP2R3A* gene were designed using the online CRISPR Design Tool provided by the Zhang Lab (http://www.genome-engineering.org/crispr/?page_id = 41). The sequences used are displayed in Fig. 6a. Oligonucleotides (Sigma) were annealed and ligated into the vector. HCT116 cells were co-transfected with both plasmids (0.45 µg each) and a plasmid carrying puromycin resistance (0.1 µg) using the transfection reagent TurboFect (ThermoFisher Scientific) according to the manufacturer's protocol. As control cells, HCT116 cell cultures received the pX330 vector without gRNA. Cells were cultivated for a week under antibiotic selection. We obtained several clones and further cultivated them without puromycin for another 7 days. Several clones devoid of PR130 and a control clone with an unaltered expression of PR130 were acquired. We analysed two PR130 null HCT116 cell clones (HCT116^ΔPR130; clone #3 and #16) and CRISPR-Cas9 control cells (HCT116^ΔgDNA). Analysis of chromosomal integrity in metaphase spreads revealed that these cell cultures had comparable chromosome numbers and no gross abnormalities.

**Fig. 7** PR130 modulates cell cycle control during replicative stress. **a** HCT116^ΔgRNA and HCT116^ΔPR130 (clone #3) cells were treated with 2 µM MS-275 and/or 1 mM hydroxyurea (HU) for 24 h. Indicated proteins and their phosphorylation were analysed by western blot; β-Actin served as loading control. **b** Densitometric analysis of CHK1 phosphorylation (S317) and total CHK1 levels after 24 h following protein detection via western blot. Data were normalised to the respective loading controls. The amount of (phosphorylated) proteins was normalised to that of HU-treated cells. Results represent the mean ± SD ($n = 8$; two-way ANOVA, ****$P < 0.0001$). **c** HCT116^ΔgRNA (left panel) and HCT116^ΔPR130 (clone #3; right panel) cells were treated with 1 mM HU and 2–5 µM MS-275 for 28 h. Cells were stained with PI and analysed using flow cytometry. Data are mean ± SD ($n = 3$). **d** Cells were incubated as stated in **a**. Immunoblots were carried out for WEE1, pHDAC2, pCDK1 and HSP90 as loading control. Numbers indicate densitometric analysis of western blot signals relative to untreated HCT116^ΔgRNA cells and normalised to HSP90 ($n = 3$). **e** Western blot analysis of p21 and β-actin (loading control). Signals of p21 are displayed relative to untreated controls and normalised to β-actin. **f** Increasing doses of 1–5 mM HU stall the cell cycle of HCT116 cells more pronouncedly in early G1/S phase. HU was applied for 24 h ($n = 3$). **g** HU (1–5 mM) was applied to HCT116 cells for 24 h. Western blot was done as indicated. Numbers indicate densitometric analysis of pCHK1, p21 and pCDK1 signals relative to untreated control and normalised to loading control ($n = 3$)

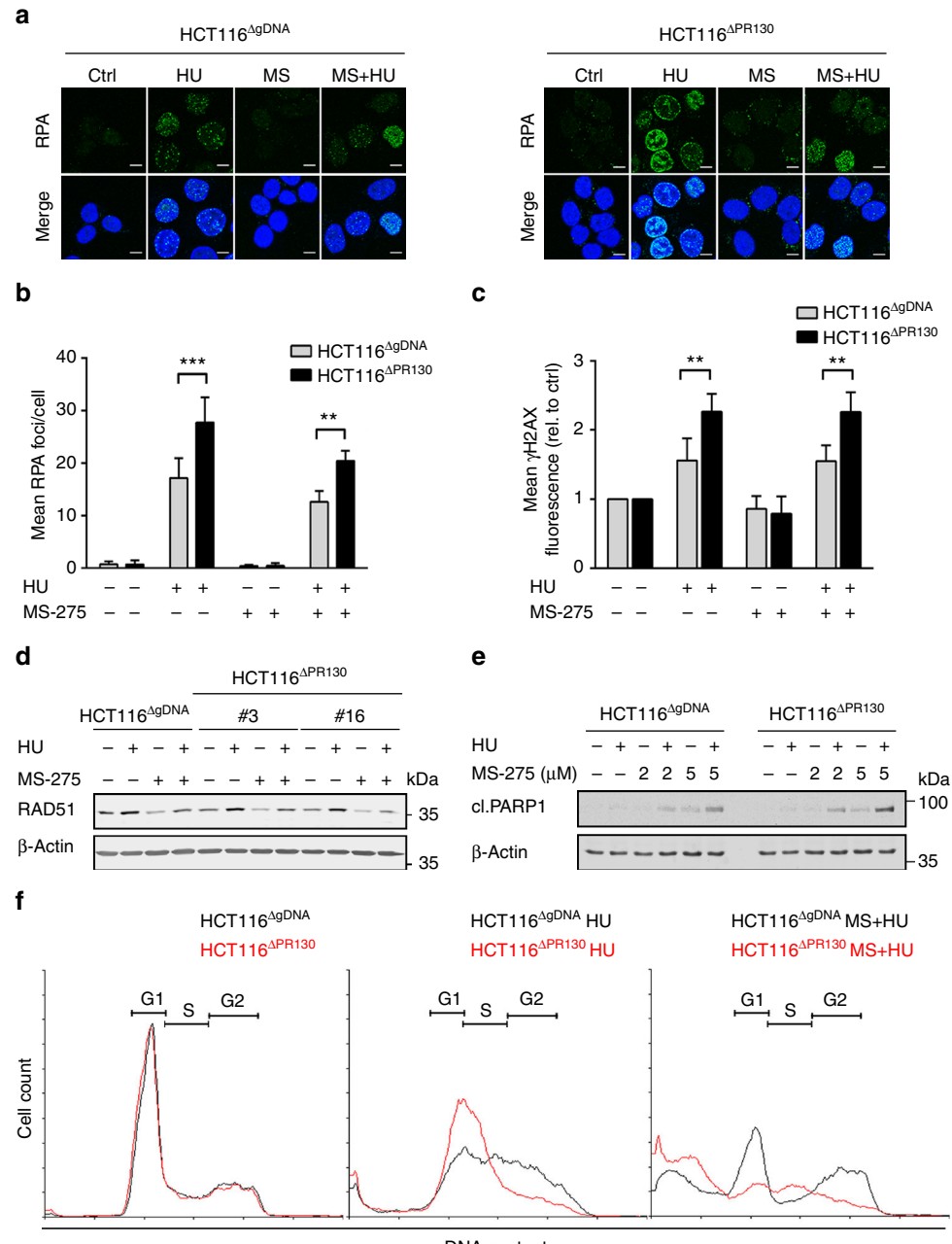

**Fig. 8** PR130 influences cell fate decisions. **a** HCT116^ΔgRNA (upper panel) and HCT116^ΔPR130 (clone #16; lower panel) cells were incubated with 1 mM hydroxyurea (HU) and 2 μM MS-275 for 24 h. Cells were fixed and incubated with RPA-specific primary and Alexa Fluor-488-coupled secondary antibody. TO-PRO3 was used to visualise nuclei. Representative images are shown; scale bar, 10 μm. **b** Analysis of RPA foci per cell using ImageJ software. Treatment and staining were performed as described in **a**. Data presented as mean ± SD ($n = 3$; two-way ANOVA; **$P < 0.01$; ***$P < 0.001$). **c** Quantification of mean γH2AX fluorescence per cell using ImageJ software. At least 50 nuclei per treatment were analysed. Cells were treated with MS-275/HU for 24 h. Results were normalised to untreated controls. Data represent the mean ± SD ($n = 3$; two-way ANOVA, **$P < 0.01$). **d** Western blot analysis of HCT116^ΔgRNA and two HCT116^ΔPR130 clones after 24 h treatment with 1 mM HU and/or 2 μM MS-275. RAD51 and β-Actin were detected using immunoblot. **e** Cells were treated with 1 mM HU and 2–5 μM MS-275 for 24 h. Whole-cell lysates were analysed for PARP1 cleavage (cl. PARP1) by western blot. β-Actin served as loading control. **f** Treatment with HU (1 mM) and/or MS-275 (2 μM) was carried out for 40 h in both HCT116^ΔgRNA (black) and HCT116^ΔPR130 (clone #3; red). Cell cycle analyses were performed with flow cytometry and representative histograms are shown ($n = 3$)

**Cell cycle analysis**. Cells were detached from the cell culture dish with accutase (GE Healthcare), washed once with phosphate-buffered saline (PBS; GE Healthcare; 200$g$/5 min) and fixed in 70% ice-cold EtOH. Samples were stored at -20 °C for at least one night. Prior to staining, EtOH was removed and cells were washed twice with PBS (200$g$/5 min). Then, they were resuspended in 334 μl PBS solution containing 12.5 μg/ml propidium iodide and 100 μg/ml RNase A (Carl Roth). To degrade all RNA, samples were incubated at 37 °C for 30 min. Cell cycle profiles were measured with FACSCanto Flow Cytometer and analysed with FACSDiva Software (BD Biosciences) and FlowJo (TreeStar Inc.) or Flowing Software (Turku Centre for Biotechnology).

**Determination of apoptosis**. Apoptotic cell death was determined with Annexin-V staining by flow cytometry analysis. Therefore, adherent cells were detached from the cell culture dish with accutase (GE Healthcare) washed with PBS (GE Healthcare; 300$g$/5 min) and resuspended in Annexin-V Binding buffer (10 mM

HEPES pH 7.4; 140 mM NaCl; 2.5 mM CaCl$_2$) containing 1 : 160 FITC-conjugated Annexin-V. After incubation for 15 min at room temperature, fluorescence intensity was measured immediately with FACSCanto Flow Cytometer and FACSDiVa Software (BD Biosciences).

**Protein lysates, western blot and densitometric analysis.** To generate whole-cell lysates, cells were harvested in 1 ml PBS on ice, combined with their supernatants and centrifuged at 300g/5 min. Pellets were washed with ice-cold PBS (300g/5 min). Cells were lysed in NET-N buffer (100 mM NaCl, 10 mM Tris-HCl pH 8, 1 mM EDTA, 10% glycerine, 0.5% NP-40; plus cOmplete protease inhibitor Tablets (Roche) and phosphatase inhibitor cocktail 2 (Sigma)) for 30 min on ice, sonicated (10 s/20% amplitude) and centrifuged (18,800g/15 min/4 °C). Protein concentrations of resulting lysates were measured by Bradford assay. Detection of proteins was performed by SDS-PAGE and western blots using enhanced chemoluminescence and X-ray films (GE Healthcare) or Odyssey Infrared Imaging System (Licor). To perform densitometric analyses, ImageJ and Image Studio Lite V4.0 (Licor) were used. Uncropped western blot images of data shown in Figs. 1–8 can be found in Supplementary Figures 7–12.

**Co-immunoprecipitation.** Cells were treated with phosphatase inhibitor (25 nM Okadaic acid) for 4 h prior to lysis. To stabilise the transient interaction of PP2A with its targets, the cells were fixed with 1% paraformaldehyde (in PBS) for 10 min followed by 5 min incubation with quenching solution (1 M glycine in PBS). After harvesting, the cells were incubated in 1 ml Co-IP lysis buffer (50 mM Tris pH 8.0, 120 mM NaCl, 0.5% NP-40; plus cOmplete protease inhibitor Tablets and phosphatase inhibitor cocktail 2) for 30 min at 4 °C followed by sonification (5 s, 20% amplitude) and centrifugation (18,800g/10 min/4 °C). An aliquot of 900 μl of lysate was rocked with protein A/G sepharose beads (Sigma-Aldrich) for pre-clearing. Supernatant was incubated with 0.6 μg pATM antibody (Abcam), or as control with same amount of IgG from rabbit (pre-immune serum, pre) or no antibody for 2 h at 4 °C. After addition of 50 μl protein A/G sepharose beads (50% slurry) rotation was continued overnight. Beads were washed five times with washing buffer (20 mM Tris pH 8.0, 1 mM EDTA, 900 mM NaCl, 0.5% NP-40) and once with final washing buffer (20 mM Tris pH 8.0, 1 mM EDTA, 100 mM NaCl, 0.5% NP-40). Proteins bound to beads were released by adding 2 × Laemmli buffer and heating the samples (95 °C, 5 min). Immunoprecipitation of PR130 was performed using 2.4 μg antibody (sc-6115) or equal amounts of goat IgG (Santa Cruz; sc-2028) and 750 μg protein. To detect acetylated-Lys 2 μM MS-275 was added to the lysis buffer. ATM-Immunoprecipitation was performed without PFA-fixation, according to ref.[24]. Cells were lysed in NET-N buffer for 1 h on ice and centrifuged at 18,800g/10 min. An amount of 2 mg protein and 0.8 μg ATM antibody (Millipore; PC116) or rabbit IgG (Santa Cruz; sc-2027) were incubated overnight at 4 °C. A/G sepharose beads were added for 4 h as described above followed by four washing steps with NET-N buffer. Proteins were extracted as described for pATM-immunoprecipitation.

**DNA fibre assay.** Exponentially growing cells were pulse-labelled with 25 μM 5-chloro-2'-deoxycytidine (CldU, Sigma-Aldrich) for 20 min while still in the presence of hydroxyurea and/or MS-275. After removing both the inhibitors and CldU, a second pulse labelling with 250 μM 5-Iodo-2'-deoxycytidine (IdU, TCI Deutschland) for 20 min was performed. The cells were harvested, lysed and DNA fibre spreads were prepared. In the following, acid-treated fibre spreads were incubated with rat anti-BrdU (Oxford Biotechnologies) that recognises CldU and subsequently mouse anti-BrdU (BD) against IdU. Goat anti-rat Cy3-coupled (Jackson ImmunoResearch, 112-166-062; 1 : 500) and goat anti-mouse Alexa Fluor-488 (Life Technologies, A-11017; 1 : 500) were used to detect the primary antibodies. Fibres were examined and pictures taken using confocal microscopy with a Zeiss Axio Observer.Z1 microscope equipped with a LSM710 laser-scanning unit (Zeiss). CldU and IdU tracks were measured using the overlay function in the LSM Image Browser (Zeiss) and micrometre values were converted into kilo bases. At least 150 forks were analysed from three repetitions. DNA fibre structures of three independent experiments were counted using ImageJ.

**Immunofluorescence.** Cells were grown on coverslips and treated as indicated. Following treatment, cells were fixed and permeabilized using ice-cold (−20 °C) methanol:acetone (7 : 3) for 10 min. The samples were washed three times with PBS and incubated with blocking solution (10% normal goat serum (NGS), 0.1% Triton X-100 in PBS) for 1 h. Cells were incubated with primary antibody (diluted in blocking solution) for 1 h at room temperature or overnight at 4 °C, followed by washing three times with PBS. For γH2AX staining, slides were washed two times and incubated for 1 min with high-salt PBS (0.4 M NaCl in PBS) followed by another washing step with PBS. Then, coverslips were incubated with the appropriate secondary antibody (in 0.1% Triton X-100 in PBS) for 1 h at room temperature. Following three washing steps (PBS) nuclear staining was performed for 15 min using TO-PRO-3 (Life Technologies) and the slides were mounted with Vectashield® (Vector Labs). Samples were analysed and images captured using confocal microscopy with a Zeiss Axio Observer.Z1 microscope equipped with a LSM710 laser-scanning unit (Zeiss).

**Neutral comet assay.** Microscopy slides were coated with 1.5% agarose in PBS. A volume of 10 μl of a cell suspension (from 1 × 10$^6$ HCT116 cells/ml) were diluted in 120 μl of 0.5% low-melting-point agarose. This suspension was added to the slide, which was then covered and incubated for 5 min at 4 °C. After removal of coverslips, the slides were incubated for 1 h in pre-cooled lysis buffer (2.5 M NaCl, 100 mM EDTA, 10 mM Tris, 1% sodium lauroyl sarcosinate; pH 7.5) at 4 °C. Electrophoresis was carried out in electrophoresis buffer (1 M Tris, 1 M boric acid, 0.5 M EDTA) to allow comet formation (25 V, 4 °C, 15 min). Slides were then fixed for 20 min in ethanol and dried overnight at room temperature. Fifty microlitres of a 50 μg/ml solution of PI was added and the slides were then analysed microscopically.

**siRNAs and transfections.** Knockdown of HDAC1, HDAC2, HDAC3 and PR72/PR130 in HCT116 cells were performed by transfecting 20 pmol siHDAC1 (Life Technologies, #4390824), 40 pmol siHDAC2 (Santa Cruz, sc-29346), 40 pmol siHDAC3 (Life Technologies, s16878), 20–40 pmol siPR72/PR130 (Life Technologies, s10975) or corresponding amounts of non-targeting control siRNA (Eurofins MWG Operon; Santa Cruz) with Lipofectamine®2000 or Lipofectamine® RNAi-MAX (manufacturer's protocol; Invitrogen, Darmstadt, Germany). Knockdown of ATM and ATR was performed using 40–50 pmol SMARTpool: siGENOME siRNA (Dharmacon, ATM: M-003201-04; ATR: M-003202-05). For simultaneous knockdown of two HDACs, different amounts of the respective siRNAs were combined as follows: 10 pmol siHDAC1, 20 pmol siHDAC2 and/or 20 pmol siHDAC3. For the combined knockdown of all three HDACs, 12 pmol siHDAC1, 24 pmol siHDAC2 and 24 pmol siHDAC3 were used. The volume of Lipofectamine® and the amount of control siRNA (siCtrl) was adjusted to constant levels. After 24–48 h, transfection mixture was removed and cells were stimulated as indicated in the Figure Legends. Efficient knockdown was confirmed by western blotting.

Plasmid transfections with pcDNA3.1 and HA-tagged PR130 were performed using TurboFect (ThermoFisher Scientific) according to the manufacturer's instructions. Forty-eight hours following transfection, MS-275 and/or hydroxyurea were added to the cells.

**In vitro PP2A phosphatase assay.** HCT116 cells were transfected with HA-tagged PR130 or pcDNA3.1 (vector control) using TurboFect (ThermoFisher Scientific). Twenty-four hours after transfection, cells were treated with 2 μM MS-275 for an additional 24 h. After harvesting, the cells were lysed for 1 h in IP lysis buffer (200 mM HEPES pH 7.4, 150 mM NaCl, 1 mM EDTA, 0.5% NP-40; plus cOmplete protease inhibitor Tablets) without the addition of phosphatase inhibitors. Immunoprecipitation of HA-tagged PR130 was performed using 3.5 μg HA-probe antibody (Santa Cruz) or equal amount of mouse IgG (Santa Cruz; sc-2025). After 2 h at 4 °C, 30 μl Protein G-Sepharose (GE Healthcare) were added for 3 h. The beads were washed once with IP lysis buffer and three times with TBS (pH 7.4). Immunoprecipitated phosphatase complexes were incubated with 750 μM phosphopeptide (Threonine phosphopeptide, #12–219, Millipore, as positive control; pATM S1981 peptide, see Peptide Synthesis and Purification section) in phosphatase reaction buffer (50 mM Tris-HCl, pH 7.0, 100 μM CaCl$_2$) for 30 min at 37 °C. The presence of free phosphate was detected 15 min after addition of Malachite green solution (Cell Signalling) by measuring the absorbance at 650 nm using Tecan Sunrise microplate reader.

**Peptide synthesis and purification.** Solid-phase synthesis of peptides was performed using an automated peptide synthesiser ResPep SL from Intavis Bioanalytical Instruments AG (Cologne, Germany) and a standard 9-fluorenylmethyloxycarbonyl (Fmoc) protocol. Rink amide MBHA resin was used as polymer. The side chain of the phosphorylated serine residue was protected as follows: Fmoc-Ser(PO(OBzl)OH)-OH. Coupling reactions were performed in DMF using Fmoc amino acids (4 equiv) activated with HBTU (4 equiv) in the presence of DIEA (8 equiv) for 0.5–1 h (double couplings). Fmoc-phospho amino acid was coupled in twofold excess with HBTU (2 equiv) and DIEA (6 equiv). Fmoc removal was effected by treating the resin twice with 20% piperidine in DMF. Acetylation was carried out using a mixture of acetic anhydride/N-methylimidazole/DMF (1 : 2 : 3) for 45 min, followed by intensive washings employing DMF (4×), DCM (3×) and repetition of these steps. Cleavage of the peptides from the resin with concomitant side-chain deprotection was achieved by treating the resins with TFA/water/triisopropylsilane (95 : 2.5 : 2.5) for 3 h. The crude peptides were precipitated in diethyl ether, centrifuged and washed three times with diethyl ether. Finally, the peptides were purified on a semipreparative reversed-phase HPLC using a Shimadzu LC-8A system (Shimadzu, Duisburg, Germany) equipped with a C18 column (Eurospher 100; Knauer, Berlin, Germany). The identity and purity of the final products were established by analytical reversed-phase HPLC (LC-10AT; Shimadzu) using a Vydac 218TP column (5 μm particle size, 300 A pore size, 4.6 × 25 mm), a gradient of 0–40% of eluent B (0.1% TFA in acetonitrile, A: 0.1% TFA in water) and detection at 220 nm. An LC-ESI mass spectrometer micrOTOF Q III (Bruker Daltonics GmbH, Germany) was used to detect the correct molar mass of the synthesis products. Analytical characterisation of the peptides revealed >98% purity. Molar masses (monoisotopic) were determined as $[M + H^+]$ for the peptide Ac-AFEEG(pS)QSTTI-amide (1290.51 g/mol, calc. 1289.41).

**Quantitative RT-PCR.** Total cellular RNA was isolated by Trizol extraction with peqGOLD RNAPureTM according to the manufacturer's protocol. The amount

and quality of the isolated RNA was analysed with the help of a NanoDrop 1000 spectrophotometer. A total of 2 µg RNA was transcribed to cDNA by means of the First Strand cDNA Synthesis Kit (Fermentas, now ThermoFisher Scientific), using random hexamer primers. qPCR was performed in triplicates with the SYBR® Green Real-Time PCR Master Mix on the StepOnePlus™ Real-Time PCR System (both Applied Biosystems). Obtained data were analysed with the $\Delta C_q$ quantification method to determine expression levels, which were normalised to two reference genes (HMBS and GAPDH). Housekeeping genes were verified with the geNorm programme. All primer sequences are listed below.

**Chromatin immunoprecipitation.** ChIP was carried out with HCT116 cells that had been treated with hydroxyurea and MS-275 for 24 h. Cellular genomic DNA and proteins were crosslinked by addition of 270 µl 37% formaldehyde to the medium (10 ml) for 10 min under constant shaking. The reaction was stopped by addition of 1 ml 1.25 M glycine. After 10 min incubation, medium was removed, cells were washed twice with PBS, collected and resuspended in 1 ml PBS containing 10 ml cell lysis buffer (50 mM HEPES, 85 mM KCl, 0.5% NP-40). After 10 min on ice, nuclei were collected by centrifugation (100g, 10 min, 4 °C) and resuspended in 500 µl nuclei lysis buffer (50 mM Tris-HCl pH 8.0, 10 mM EDTA, 1% SDS, protease Inhibitors). After 10 min incubation on ice, genomic DNA was fragmented by sonication (six times for 10 s at output 6) to a fragment size of ~500 bp, using a Branson Sonifier 250. Thereafter, SDS was removed by centrifugation (18,800g, 15 min, 4 °C) and DNA concentration was measured using NanoDrop™ 1000. Equal amounts of fragmented DNA were filled with dilution buffer (16.7 mM Tris-HCl pH 8.0, 1.2 mM EDTA, 167 mM NaCl, 1% SDS, protease Inhibitors) up to 800 µl and protein G sepharose was added for pre-clearing. After incubation for 1 h at 4 °C, protein G sepharose was removed by centrifugation (900g, 5 min, 4 °C); specific antibodies (see above, section Antibodies) were added and incubated overnight at 4 °C. Thereafter, protein G-sepharose was added for 2 h at 4 °C. After centrifugation (900g, 5 min, 4 °C) the immunoprecipitate was washed three times in 1 ml high-salt wash buffer (50 mM HEPES, 1% Triton X-100, 0.1% SDS, 500 mM NaCl, 1 mM EDTA, 0.1% Deoxycholate) and two times TE8 buffer (10 mM Tris, 1 mM EDTA pH 8.0). After the final washing step, the immunoprecipitate was resuspended in 300 µl elution buffer (50 mM Tris-HCl pH 8.0, 10 mM EDTA, 1% SDS, protease Inhibitors) + 1 µl proteinase K (25 mg/ml) and incubated 2 h at 55 °C, followed by overnight incubation at 65 °C. DNA was isolated by phenol–chloroform extraction, recovered by ethanol/NH4Ac precipitation and resuspended in aqua bidest. qPCR was performed using primers amplifying a 196 bp fragment located 422–226 bp upstream of the first exon of hPPP2R3A (see below, Primer sequences for RT-pPCR and ChIP).

**Primer sequences for RT-qPCR and ChIP.** Following primer sequences were used for RT-qPCR: PPP2R3A: 5′-GCGGTCCAGAAGGATGTTGAGA-3′ (forward), 5′-TCCTGGAATTGTGCTTGGGC-3′ (reverse); GAPDH: 5′-TGCACCAC-CAACTGCTTAGC-3′ (forward), 5′-GGCATGGACTGTGGTCATGAG-3′ (reverse); HMBS: 5′-GGCAATGCGGCTGCAA-3′ (forward), 5′-GGGTACC-CACGCGAATCAC-3′ (reverse). To perform ChIP analyses, the following primers were used: PPP2R3A: 5′-TGAGTGAAATGTGGCGCAGG-3′ (forward), 5′-TCGAGTCTCTGGCTCGGATTT-3′ (reverse).

**Microarray analysis.** Total cellular RNA was extracted and its quality assessed as described in 'Quantitative RT-PCR'. RNA processing and microarray analysis itself were performed at the European Molecular Biology Laboratory (EMBL) in Heidelberg. GeneChip® Human gene 2.0 ST Array (Affymetrix) was used for transcriptome analysis.

**Statistical analysis.** Statistical analysis was carried out with Student's t-test, one-way ANOVA and two-way ANOVA as indicated. Statistical analysis was corrected for multiple testing using Tukey's multiple comparisons test to calculate the SD for independent experiments. Respective P values as a measure of significance are indicated.

**Data availability.** The data that support the findings of this study are available from the corresponding author upon reasonable request. Microarray data are deposited at NCBI Geo, accession number 'GSE108868'.

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

## Acknowledgements

We thank all members of our groups for helpful discussion, and we thank S. Reichardt, T. Brachetti and S. Scheiding for technical assistance. This work was supported by grants from the Deutsche Krebshilfe (Grant No. FKZ102362 (110908/110909) to G.S. and O.H.K.), the Wilhelm Sander-Stiftung (Grant No. 2010.078.1 to O.H.K.) and the Deutsche Forschungsgemeinschaft (Grant Nos. KR2291/5-1 and KR2291/7-1 to O.H.K.). HCT116 cells were kindly provided by Professor Dr. B. Vogelstein (Baltimore, MD, USA). RKO and K562 cells were kindly provided by Dr. M. Zörnig and Dr. M. Grez (both Frankfurt/Main, Germany), respectively. HA-PR130 plasmid was a kind gift from Professor Dr. R. Bernards (Amsterdam, the Netherlands). We thank Dr. V. Benes (Heidelberg, Germany) and Dr. M. Wirth (Munich, Germany) for help with microarray analysis and Professor Dr. J. Fahrer (Gießen, Germany) for helpful discussions.

## Author contributions

A.G., C.E., T.N., N.K., T.K., M.C., and M.S. performed laboratory work. All authors analysed data. T.H., G.S., D.I., and O.H.K. provided material. A.G., C.E., G.S., and O.H.K. wrote the paper. O.H.K. initiated the project.

## Additional information

**Competing interests:** The authors declare no competing interests.

