## [Peer Review File · Nature Communications]

Reviewers' Comments:

Reviewer #1 (Remarks to the Author)

This study is a coherent and easy to follow study as it progresses stepwise upon the data achieved and logical assessment. Essentially it concludes that PR130 is a novel regulator of cell fate decisions upon replicative stress, as HDAC1 and 2 inhibit PR130 subunit of PP2A and promotes phosphorylation of ATM and CHK1 by ATR.

I would like to point the following:

1 In the absence of PR130 (by CRISPER-Cas9, or by siRNA) how were the HDACS 1 and 2 affected directly, if at all?

2 Why the attenuation of CHK1 phosphorylation by HDACi occurred in both PR130-positive and -negative cells? It raises a question about the relative role of PR130 as probably other subunits are involved? Are the authors able to show the change in the other B subunits within the CRISPER-Cas9 silent PR130 cells?

3 In the literature several mechanisms were described to explain how cells can re-enter the cell cycle even when checkpoint kinase signaling is lost. However, it is not clear to me how it is explained in their cancer-derived systems of this manuscript.

Minor:

1 In the results, the authors describe that within 6 hours, MS-275 did not alter the HU-induced phosphorylation of ATM, however, in the Figure it is shown that it did decrease pATM in 6h and 24h

2 Why in Figure 1e in the control, there is a decrease in pCHK1 as time after UV prolongs? (the opposite is with pCHK2 and pATM and pATR)

3 The authors describe that MS-275 led to a significant accumulation of cells in G2 phase (Fig. 2a-b), but it looks there is an accumulation in G1 phase (70%) and not in G2 (25%)

4 They used okadaic acid and cantharidin as PP2A inhibitors, but the authors should check if the doses used are relevant for the specific inhibition of PP2A

5 Please use, at least in the beginning the subunits more official name PPP2R3A.

6 Another subunit Bbeta (Proc Natl Acad Sci U S A. 2011 Jul 26;108(30):12443-8) controls IL-2 deprivation cell death. I wonder if the authors can see a possible connection.

Reviewer #2 (Remarks to the Author)

In an intriguing paper, Goder et al report that phosphatase PP2A subunit PR130 expression is regulated by the histone deacetylase HDAC1/HDAC2. Moreover, PR130 controls the phosphorylation status of ATM, CHK1 and CHK2 but not ATR in response to replicative stress. As a result, HDAC1 inhibitors, MS275 mediates dephosphorylation of ATM/CHK1 and when combined with replicative stress such as that induced by RRM2 inhibition by hydroxyl urea, can result mitotic catastrophe followed by apoptosis.

Overall comments:

The authors uncover for the first time that PR130 subunit of PP2A is required for replicative stress response induced DNA damage. The paper is characterized by a degree of novelty. The authors convincingly demonstrate using RNA silencing, genetic ablation and pharmacological blockade that HDAC1/PR130 circuit adds another layer of regulatory control in cellular response to replicative stress. By regulating the phosphorylation of ATR targets, CHK1 and the ATM/CHK2 signaling at later times after initial induction of replicative stress with hydroxyl urea i.e 6h versus 24 hours, the histone deacetylase ensures a backup mechanism of maintaining genome integrity in response to replicative stress.

While the authors demonstrate that simultaneous knockdown of HDAC1 and HDAC2 increased expression of PR130. The additional elimination of HDAC3 had no effect on PR130 (Fig. 5e and

Supplementary Fig. S3b-c). A combined reduction of HDAC1 and HDAC2 impaired the HU-induced phosphorylation of ATM and CHK1 (Fig. 5f and Supplementary Fig. S3d).

The authors demonstrate universality of the mechanism by using three different cell lines; the colon cancer cells, HCT116 and RKO colon cancer cells as well as murine embryonic fibroblasts and three different replicative stress response agents, hydroxy urea (HU), ultraviolet light (UV) and 5 fluorouracil (5-FU).

Major points

1. There are no CHIP experiments to support recruitment of HDAC1/2 to PR130 promoter and inhibition of this recruitment by MS-275.
2. Is there a role for PR130 in IR or etoposide induced activation of ATM/pCHK2/p53 G2/M checkpoint? What is the effect of MS-275 on IR-induced pATM/CHK2/WEE1 mediated G2/M checkpoint? Is it specific only to replicative stress?
3. As HDACs inhibitors not only regulate gene expression but also protein acetylation, the authors have to demonstrate whether there is a direct interaction between PR130 and CHK1 or CHK2 in response to HU by co-immunoprecipitation experiments.
4. The paper is confusing to read at times, particularly the role of pCHK1 in the time delayed response is not convincing! The authors need to add more into discussion explaining why the benzamide MS-275, which specifically inhibits HDAC1,-2,-3 is specifically inhibiting CHK1 phosphorylation without affecting its upstream kinase ATR.

An alternative explanation for the results observed could be that, if unresolved replication forks are leading to double strand breaks in a time delayed manner leading to the activation of ATM/pCHK2 in a time-delayed manner, this may be sufficient to induce a WEE1 mediated G2/M arrest. The result suggests that HDAC1/2/3 mediated PR130 suppression may be activated following initial pATR/pCHK1 mediated replicative stress response to resolve any unrepaired breaks. pATM/pCHK2/WEE1 signaling may then be required for the subsequent G2/M arrest.

This would be consistent with the authors observation that "MS-275 reduces HU-induced RPA foci (active ATR mediated intra-S phase checkpoints), while γ H2AX foci persist (Fig. 3c-d).

5. Furthermore, ATM inhibition increases HU-induced γ H2AX indicating DSBs (Fig. 4b). ATM is the kinase for phosphorylating H2AX at serine 139 and indeed the authors report that compared to isogenic wild-type cells, CHK2^{-/-} cells show less PARP1 cleavage and reduced γ H2AX indicative of a functional ATM/pCHK2 response to HU treatment (Fig. 4i). It is unclear, as to how does its inhibition increase HU- induced DSB promoted H2AX phosphorylation. If the authors are invoking ATR as the kinase, they have to provide experimental evidence that in these cells if ATR is silenced by RNAi, H2AX phosphorylation is completely abrogated. Moreover, Fig 4b and 4d clearly shows that HU-induced pCHK1 phosphorylation is independent of ATM.

6. Genetic experiments using PR130 deletion cell lines suggest that PR130 mediated regulation of replicative stress response cannot be attributed directly to changes in CHK1 phosphorylation. For example, while the HU- 210 induced phosphorylation of ATM remained elevated despite the presence of MS-275 (Fig. 6b). In contrast, MS-275 appears to decrease CHK1 phosphorylation in PR130 -negative cells and PR130 positive cells (Fig. 6c-d), which would also suggest that PR130 is dispensable for CHK1 phosphorylation, as in PR130 negative cells, there is no phosphatase subunit and CHK1 phosphorylation if it is PR130 dependent should be intact regardless of MS-275 addition!

Thus, two distinct molecular mechanism appear to be operational here. The paper suffers from the authors inability to separate these two events.

Reviewer #3 (Remarks to the Author)

Goeder et al. investigated the role of histone deacetylases (HDACs) 1-3 in DNA replication checkpoint regulation, mostly in the epithelial colon cancer cell line HCT116. Long-term replication stress concomitant with HDAC inhibition decreased phosphorylation of the ATM and CHK1+2 kinases at key regulatory sites and this decreased checkpoint kinase phosphorylation correlated with checkpoint slippage and cell death. The decrease in phosphorylation pointed to the involvement of a phosphatase and indeed by a transcriptome profiling approach the authors identified a regulatory subunit of protein phosphatase 2A (PP2A), PR130, as a potential candidate responsible for the checkpoint kinase dephosphorylation. Consistent with their hypothesis deletion of PR130 or siRNA driven knock-down of PR130 abrogated the decrease in ATM phosphorylation upon replication stress and HDAC inhibition. CHK1, on the other hand, still got dephosphorylated to the same extent as in wt PR130 cells suggesting the involvement of another phosphatase. However, the application of replication stress alone caused a significant increase of CHK1 phosphorylation in cells lacking PR130 still suggesting a role for PR130 in this process. Consistent with the enhanced CHK1 phosphorylation in the delta PR130 cells these cells arrested earlier than wt cells in G1/S phase. Upon replication stress these cells showed an increased number of RPA foci compared to wt cells and additional HDAC inhibition led to a decrease in RPA foci (indicating ssDNA) in both cell types. γ H2AX foci indicating double strand breaks were also increased in delta PR130 cells upon replication stress but additional HDAC inhibition did not further increase their level. Interestingly, more delta PR130 than wt cells underwent apoptosis (from the graph in Fig.7a it is difficult to read/deduce the % of cells in subG1) when exposed to replication stress and HDAC inhibition. The authors argued that the increased cell death is due to the attenuated CHK1 phosphorylation (ATM phosphorylation was unchanged) and the attenuated RAD51 levels. In principle these are interesting findings in particular for the field of cancer therapy. A major weakness of this study is the lack of data on the mechanism underlying the observed phenomenon (PR130 regulating ATM and CHK1 phosphorylation). Moreover, induction of a 24h replication stress concomitant with a 24h inhibition of HDACs1-3, which are major epigenetic regulators of probably thousands of genes, is a very artificial condition. What is the physiological relevance of results made under such experimental conditions? What can we learn from these data about the "normal" replication checkpoint regulation by HDACs and PP2A?

For example, the authors conclude from their data that PR130-PP2A "promotes dephosphorylation" of ATM and CHK1.

The authors should test directly if the PR130-PP2A holoenzyme is able to dephosphorylate ATM (at S1981, but also at S367 and S1893) and CHK1 (S317 and S345) and compare it to other di- or trimeric PP2A holoenzymes.

Moreover, in Fig. 6F an interaction is shown between ATM and PR130. Several questions arise from this result and have to be answered:

Is the fraction of ATM that is found in complex with PR130 dephosphorylated at the aforementioned sites?

Is ATM also co-precipitating the other holoenzyme components (catalytic C and structural A subunit)? If yes, is the complex catalytically active when associated with ATM?

Is ATM only associated with PR130-holoenzymes upon replication stress and HDAC inhibition or is it already associated even in unstressed cells? If already bound what is the signal to get it activated?

The same analysis should be performed with CHK1 and PP2A.

Minor points:

Not only a subset but all Western blot data need to be quantified (from at least 3 independent experiments), see below comments to Fig.5b and 5e.

The authors should provide % for the cell cycle distribution graphs in Fig. 2a and 7a and should refer in the text to the exact numbers and phrase carefully. For example Fig.7c shows a decrease of RPA foci upon HDAC inhibition under replication stress, in delta PR130 cells there is a decrease from ~27 RPA foci/cell to ~22 and the authors wrote in line 230: "The MS-275-induced loss of RPA foci ...".

Line 178, the authors wrote that PP2A inhibition by cantharidin rescued the inhibition of checkpoint kinase phosphorylation. However, in Fig.5b there is still a substantial decrease in pATM and (even more) in pCHK1 phosphorylation levels visible despite the inhibition of PP2A. Therefore, as asked for above, all western blot data need to be quantified and secondly, the authors need to phrase more carefully their conclusions.

Line 192 + Fig 5e, once again, the authors claim based on a single Western blot experiment that knockdown of HDAC1+2 would lead to increased expression of PR130. However, this is not evident from the single experiment shown in Fig.5e. This experiment needs to be repeated and the signals need to be quantified.

We would like to thank all reviewers for the constructive and thoughtful comments on our work and for recognizing the novelty and impact of our results. We addressed all issues experimentally and we strongly believe that our new data significantly improve the quality of our article. Below we briefly sum up the key novel findings of our revised manuscript. We rewrote the manuscript to make it easier to read and we provide point-by-point responses to the reviewers' comments and a list summarizing new and revised figures.

The most important new data that we present are:

- The histone deacetylases HDAC1 and HDAC2 bind to the PR130 promoter *in vivo*.
- HDAC inhibition with MS-275 specifically diminishes the phosphorylation of ATM during replicative stress.
- MS-275 triggers the interaction between PR130 and phosphorylated ATM.
- PR130 is acetylated and MS-275 increases its acetylation *in vivo*.
- MS-275 increases the mRNA and protein expression of PR130, but other PP2A B subunits are not affected at the mRNA or protein levels.
- Immunoprecipitates of PR130 from cells treated with MS-275 contain the catalytic and structural PP2A subunits and such complexes are active against the phosphorylated ATM peptide sequence *in vitro*.
- There is unscheduled origin firing and cells undergo a slippage from S phase into mitotic catastrophe when hydroxyurea and MS-275 are applied together. These processes can be explained by a reduction of the cell cycle regulatory kinase WEE1 and its negative impact on the cyclin-dependent kinase CDK1. In addition, MS-275 attenuates the phosphorylation and activity of the tumor suppressor p53, which halts cell cycle progression upon replicative stress.
- We can now distinguish between two distinct novel molecular mechanisms that regulate the dephosphorylation of ATM and CHK1. We reveal that PR130 controls the dephosphorylation of ATM in conjunction with the PP2A phosphatase directly and we identify that PR130 regulates a cell cycle-dependent control of CHK1 phosphorylation. In comparison with PR130 wild-type cells PR130 negative cells progress much slower from G1 phase into S phase upon replicative stress. Nonetheless, these lower numbers of cycling cells show a higher phosphorylation of CHK1, reduced WEE1/pCDK1 signaling, and increased replicative stress with higher amounts of RPA foci, phosphorylated H2AX, and p53. When we applied increasing doses of hydroxyurea, we could verify that an earlier arrest in S phase leads to a higher phosphorylation of CHK1. HDAC inhibition with MS-275 interferes with the pronounced arrest of PR130 null cells, causes dephosphorylation of CHK1, and apoptosis. Thus, the HDAC1/HDAC2-dependent regulation of PR130 is a newly

identified regulator of checkpoint kinase phosphorylation as well as a novel upstream regulator of cell cycle progression, replicative stress, DNA damage, and apoptosis.

New and revised Figures:

1. Figure 2f
Regulation of WEE1 and CDK1 by hydroxyurea and MS-275
NEW
2. Figure 2g
Regulation of Cyclin B by hydroxyurea and MS-275
NEW
3. Figure 2h
Regulation of p53 and its phosphorylation by hydroxyurea and MS-275
NEW
4. Figure 2i
Regulation of p53 target genes by hydroxyurea and MS-275
NEW
5. Figure 5g
Data from ChIP analyses verifying the interaction of HDAC1 and HDAC2 with the PR130 promoter
NEW
6. Figure 6d
Interaction between phosphorylated ATM and PR130
Extended figure with NEW data
7. Figure 6e
Acetylation of PR130
NEW
8. Figure 6f
Dephosphorylation of a phosphorylated ATM peptide sequence by PR130
NEW
9. Figure 7d
Regulation of WEE1, CDK1, and pHDAC2 by hydroxyurea and MS-275 in HCT116^{ΔgRNA} and HCT116^{ΔPR130} cells
NEW
10. Figure 7e
Dose-dependent control of cell cycle progression by hydroxyurea
NEW
11. Figure 7f

- Dose-dependent control of CHK1 phosphorylation by hydroxyurea
NEW
12. Supplementary Figure S2b
Comet assay detecting DSBs
NEW
13. Supplementary Figure S3a
Shorter exposure for pCHK1
Revised
14. Supplementary Figure S4b
Expression of HDAC3 in HCT116 Δ gRNA and HCT116 Δ PR130 cells
Extended
15. Supplementary Fig. S4c
Expression of B56 β and PR48 proteins in the presence of hydroxyurea and MS-275
NEW
16. Figure S4d
Interaction of PR130 with PP2A-A and -C
NEW
17. Figure S4e
Response of cells to γ -IRR in the presence of MS-275
NEW
18. As requested by Reviewer 3, we have quantified all key experiments and provide the signal intensities in our revised manuscript.
NEW/Extended

Point-by-point responses to the reviewers' comments

Reviewers' comments:

Reviewer #1 (Remarks to the Author):

This study is a coherent and easy to follow study as it progresses stepwise upon the data achieved and logical assessment. Essentially it concludes that PR130 is a novel regulator of cell fate decisions upon replicative stress, as HDAC1 and 2 inhibit PR130 subunit of PP2A and promotes phosphorylation of ATM and CHK1 by ATR.

We thank the reviewer for the very positive and encouraging assessment of our work, the recognition of the novelty of our data, as well as for the critical evaluation and suggestions. We addressed all issues raised as follows.

I would like to point the following:

1 In the absence of PR130 (by CRISPER-Cas9, or by siRNA) how were the HDACS 1 and 2 affected directly, if at all?

As suggested, we analyzed the levels of the class I HDACs HDAC1 and HDAC2 in HCT116 cells with and without PR130. Due to the complete elimination of PR130 in the CRISPR-Cas9 HCT116^{ΔPR130} cells, we compared them to the corresponding HCT116^{ΔgRNA} control cells. We found no significant differences in the expression of HDAC1 and HDAC2 (**Supplementary Fig. S4b**). In addition, we determined the levels of HDAC3 and found that an elimination of PR130 does also not affect HDAC3 (**Supplementary Fig. S4b**).

Prompted by the referee's comment, we additionally assessed the phosphorylation of HDAC1 and HDAC2. While we could not detect pHDAC1, we observed that MS-275 led to an accumulation of pHDAC2. However, PR130 does not affect this increase in pHDAC2 (**Fig. 7d**).

These data are consistent with an equal induction of histone hyperacetylation in cells with or without PR130 in response to MS-275 (**Supplementary Fig. S4b**). Overall, we conclude that PR130 does not affect the expression and activities of HDAC1 and HDAC2.

2 Why the attenuation of CHK1 phosphorylation by HDACi occurred in both PR130-positive and -negative cells? It raises a question about the relative role of PR130 as probably other subunits are involved? Are the authors able to show the change in the other B subunits within the CRISPER-Cas9 silent PR130 cells?

Indeed, this is a very important question. In our original manuscript, we speculated that the dephosphorylation of CHK1 in response to MS-275 relied on PP2A. Now we present a refined and more advanced model. We are able to distinguish two distinct novel molecular mechanisms. We demonstrate a direct dephosphorylation of pATM by PR130-PP2A and a cell cycle-dependent control of CHK1 phosphorylation that is modulated by PR130.

Our data show that in response to replicative stress, PR130 null cells progress slower from G1 to S phase than corresponding PR130 positive cells. However, the lower numbers of cycling PR130 negative cells have even more replicative stress. They carry higher levels of RPA foci, pH2AX, pCHK1, and activated p53 together with a lower activity of WEE1 against

CDK1 (**Figure 7a-d** and **8a-c**). We therefore conclude that PR130 is a novel regulator of this early section of S phase that is particularly prone to replicative stress.

MS-275 reduces both pCHK1 and WEE1. Accordingly, cell cycle control is lost and PR130 null cells that are arrested in early S phase disappear and the numbers of cells in G2/M phase and apoptosis increase (**Figure 8e-f**).

We additionally show that increasing doses of hydroxyurea applied to HCT116 cells lead to an earlier arrest in S phase and to a dose-dependent increase in pCHK1 that does not require a corresponding increase in WEE1 activity (**Figure 7e-f**). These data suggest that an earlier arrest in S phase –evoked by a higher dose of hydroxyurea or a loss of PR130– leads to a more pronounced phosphorylation of CHK1.

Last but not least, our finding that the inhibition of PP2A with okadaic acid and cantharidin can rescue the hydroxyurea-induced phosphorylation of ATM, but not pCHK1 from dephosphorylation in the presence of MS-275 illustrates PP2A-dependent and -independent pathways (**Figure 5b** and **Supplementary Fig. S3a**). These data are also coherent with our observation that it is specifically the ATR kinase that catalyzes the phosphorylation of CHK1 in response to hydroxyurea (**Fig. 4d,f**) and that a rescue of ATM phosphorylation is therefore not able to increase the phosphorylation of CHK1 (**Fig. 5b**).

Regarding the other PP2A B subunits, we analyzed our microarray data for their expression. We found that MS-275 increased the mRNA levels of PR130/PPP2R3A and of B56 β /PPP2R5B (this factor was mentioned by the reviewer in her/his last comment), but no induction of other B subunits by MS-275 (**Supplementary Table 1**). Therefore, we analyzed the protein expression of this factor. As control we analyzed the levels of PR48/PPP2R3B, which was not increased in the array. We found no significant alteration of their expression in our CRISPR-Cas9 cells with and without PR130 (**Supplementary Fig. S4c**). We additionally show in this panel that the levels of the catalytic and structural subunits of PP2A are not affected by the elimination of PR130.

3 In the literature several mechanisms were described to explain how cells can re-enter the cell cycle even when checkpoint kinase signaling is lost. However, it is not clear to me how it is explained in their cancer-derived systems of this manuscript.

Due to the eminent roles of the tumor suppressor p53 and the cell cycle gatekeeper kinase WEE1 for cell cycle progression and origin firing during replicative stress, we analyzed these molecules. The slippage of hydroxyurea plus MS-275 treated cells from S phase into mitotic catastrophe can be explained by a negative regulation of WEE1. MS-275 decreases WEE1 and its negative effect on the cell cycle-promoting CDK1 (**Figure 2f, 7d** and **Supplementary**

Fig. S4e). In addition, we demonstrate that MS-275 attenuates the phosphorylation and activity of the tumor suppressor p53, which halts cell cycle progression for DNA repair upon replicative stress (**Figure 2h** and **6b**). A subsequent modulation of two p53 target genes, the cyclin-dependent kinase inhibitor p21 and the DNA repair relevant phosphatase WIP1, can also explain unscheduled cell cycle progression and DNA damage in cells treated with hydroxyurea and MS-275 (**Figure 2i** and **3e-f**).

Minor:

1 In the results, the authors describe that within 6 hours, MS-275 did not alter the HU-induced phosphorylation of ATM, however, in the Figure it is shown that it did decrease pATM in 6h and 24h

Indeed, this needs clarification. We measured and normalized the band intensities and found no significant attenuation of hydroxyurea-induced pATM after a 6h incubation period with MS-275 (**Fig. 1c**).

2 Why in Figure 1e in the control, there is a decrease in pCHK1 as time after UV prolongs ? (the opposite is with pCHK2 and pATM and pATR)

These processes are typically seen in UV-treated cells (reviewed in Chen and Sanchez, DNA Repair 2004). UV was given as a pulse and pCHK1 stalls cell cycle progression. The activation of the other checkpoint kinases indicates DNA repair processes by nucleotide excision repair and homologous recombination.

3 The authros describe that MS-275 led to a significant accumulation of cells in G2 phase (Fig. 2a-b), but it looks there is an accumulation in G1 phase (70%) and not in G2 (25%)

The referee is right, this is the case for MS-275 and we corrected this in our revised manuscript. In combination with hydroxyurea, MS-275 promotes entry into G2/M-phase and mitotic catastrophe (**Figure 2a-e**). The above-mentioned reduction of WEE1, pCHK1, and p53 activity by MS-275 are mechanistic explanations for the slippage of cells from S phase into mitotic catastrophe.

4 They used okadaic acid and cantharidin as PP2A inhibitors, but the authros should check if the doses used are relevant for the specific inhibition of PP2A

We used doses that are used throughout the literature to block PP2A specifically. For example, in a recent publication by Prof. M. Avkiran (Weeks et al., JAMA 2017) a dose of 100 nM/L okadaic acid was found to inhibit PP2A without any effect on PP1 and we used 25 nM/L okadaic acid in our experiments. These data suggest that we inhibit PP2A more efficiently than PP1. However, we agree that inhibitors may cause additional effects. Therefore, we controlled and verified all inhibitor-based experiments with genetic strategies. The resulting conclusions with RNAi and CRISPR-Cas9 against PR130 and RNAi against HDAC1 and HDAC2 are consistent with the data obtained with pharmacological inhibitors (**Figs. 6-8**).

5 Please use, at least in the beginning the subunits more official name PPP2R3A.

We entirely agree and followed this as requested (please see *Abstract* of the revised manuscript).

6 Another subunit Bbeta (Proc Natl Acad Sci U S A. 2011 Jul 26;108(30):12443-8) controls IL-2 deprivation cell death. I wonder if the authors can see a possible connection.

We discuss this publication in our revised manuscript and we tested for a putative regulation of B56 β /PPP2R5B by hydroxyurea and MS-275. We found that MS-275 increased the mRNA levels of B56 β (**Supplementary Table 1**), but not the B56 β protein levels (**Supplementary Fig. S4b**). Such data suggest that the increase in PR130 is a rather specific effect of MS-275 on B subunits.

Reviewer #2 (Remarks to the Author):

In an intriguing paper, Goder et al report that phosphatase PP2A subunit PR130 expression is regulated by the histone deacetylase HDAC1/HDAC2. Moreover, PR130 controls the phosphorylation status of ATM, CHK1 and CHK2 but not ATR in response to replicative stress. As a result, HDAC1 inhibitors, MS275 mediates dephosphorylation of ATM/CHK1 and when combined with replicative stress such as that induced by RRM2 inhibition by hydroxyl urea, can result mitotic catastrophe followed by apoptosis.

Overall comments:

The authors uncover for the first time that PR130 subunit of PP2A is required for replicative stress response induced DNA damage. The paper is characterized by a degree of novelty. The authors convincingly demonstrate using RNA silencing, genetic ablation and pharmacological blockade that HDAC1/PR130 circuit adds another layer of regulatory control in cellular response to replicative stress. By regulating the phosphorylation of ATR targets,

CHK1 and the ATM/CHK2 signaling at later times after initial induction of replicative stress with hydroxyl urea i.e 6h versus 24 hours, the histone deacetylase ensures a backup mechanism of maintaining genome integrity in response to replicative stress.

While the authors demonstrate that simultaneous knockdown of HDAC1 and HDAC2 increased expression of PR130. The additional elimination of HDAC3 had no effect on PR130 (Fig. 5e and Supplementary Fig. S3b-c). A combined reduction of HDAC1 and HDAC2 impaired the HU-induced phosphorylation of ATM and CHK1 (Fig. 5f and Supplementary Fig. S3d).

The authors demonstrate universality of the mechanism by using three different cell lines; the colon cancer cells, HCT116 and RKO colon cancer cells as well as murine embryonic fibroblasts and three different replicative stress response agents, hydroxy urea (HU), ultraviolet light (UV) and 5 fluorouracil (5-FU).

We thank the reviewer for the very positive and encouraging assessment of our work, for recognizing its novelty and impact, as well as for the critical and constructive evaluation and suggestions. We addressed all issues raised as follows.

Major points

1. There are no ChIP experiments to support recruitment of HDAC1/2 to PR130 promoter and inhibition of this recruitment by MS-275.

We agree that this needs to be shown experimentally. We carried out ChIP experiments to analyze whether HDAC1 and HDAC2 are found at the *PR130* promoter; acetylated histone H3 was used as a control to test efficient HDAC inhibition. Indeed, HDAC1 and HDAC2 can be physically detected at this promoter (**Figure 5g**). The reviewer also asked for an inhibition of HDAC recruitment by MS-275. We see a reduced localization of inhibited HDAC1 at the *PR130* promoter and the binding of HDAC2 is less affected by MS-275. We assume that this is due to an association of HDAC1 and HDAC2 with different transcription factors as both cannot bind DNA directly. Moreover, we report that MS-275 increases the phosphorylation of HDAC2 (**Fig. 7d**), which has been reported to promote its association with chromatin (Sun et al., JBC 2002). However, it is most relevant for our work and our conclusions that the efficient pharmacological inhibition of both HDACs by MS-275 permits an increased acetylation of the *PR130* promoter and the induction of PR130 (**Figs. 5 and 7**).

2. Is there a role for PR130 in IR or etoposide induced activation of ATM/pCHK2/p53 G2/M checkpoint? What is the effect of MS-275 on IR-induced pATM/CHK2/WEE1 mediated G2/M

checkpoint? Is it specific only to replicative stress?

Prof. Penny Jeggo's group has shown that ATR phosphorylates ATM in cancer cells and that this mechanism is independent of double strand DNA breaks and the MRN complex (Stiff et al., EMBO J. 2006). This mechanism differs from ATM activation by direct dsDNA damage. Others similarly found that ATM activation is different during replicative stress and laser-generated direct double strand breaks (Duquette PLOS Genetics 2012).

Nonetheless, we agree with the reviewer that it is interesting to determine whether MS-275 affects ATM phosphorylation in cells with dsDNA breaks that are induced directly by gamma-irradiation. Therefore, we irradiated HCT116 cells and analyzed pATM by Western blot. We noted that a pre-incubation with MS-275 that efficiently induced PR130 did not affect pATM when it is induced directly by gamma-irradiation (**Supplementary Fig. S4e**). We assume that this is due to an autophosphorylation of ATM and a different structural composition of the ATM complex within the MRN complex that is formed upon direct DNA damage. These data reveal that the HDAC1/HDAC2-dependent control of PR130 has a specific impact on replicative stress signaling.

When we analyzed effects of MS-275 on the gamma-irradiation-induced pATM/CHK2/WEE1 mediated G2/M checkpoint, we found that MS-275 reduced the levels of WEE1 significantly. This reduction of WEE1 by MS-275 is associated with reduced pCHK1 in gamma-irradiated cells and pCHK2 is controlled in a similar fashion (**Supplementary Fig. S4e**).

Hence, while HDAC inhibitors specifically evoke a loss of ATM phosphorylation during replicative stress, they cause a loss of CHK1 phosphorylation in response to replicative stress and direct dsDNA damage. The loss of the pATM/CHK2/WEE1 checkpoint and the associated loss of pCHK1 are coherent with our novel model in which the stress-dependent phosphorylation of CHK1 depends on cell cycle progression (please see response to Point 4).

3. As HDACs inhibitors not only regulate gene expression but also protein acetylation, the authors have to demonstrate whether there is a direct interaction between PR130 and CHK1 or CHK2 in response to HU by co-immunoprecipitation experiments.

We focused these analyses on CHK1, as we see no protective effect of CHK2 on the survival of hydroxyurea-treated HCT116 cells (**Fig. 4i**). We immunoprecipitated pCHK1 (n = 3) and total CHK1 (n = 2) and we could not detect any interaction of them with PR130 (data not shown). Therefore, we had to reconsider our conclusions and to change our model.

Please see our answer to the next point, in which we explain our novel data and the new conclusions regarding a cell cycle-dependent, previously unrecognized control of CHK1 in

response to hydroxyurea. These data and conclusions are entirely in agreement with the lack of interaction between PR130 and pCHK1.

4. The paper is confusing to read at times, particularly the role of pCHK1 in the time delayed response is not convincing! The authors need to add more into discussion explaining why the benzamide MS-275, which specifically inhibits HDAC1,-2,-3 is specifically inhibiting CHK1 phosphorylation without affecting its upstream kinase ATR.

Indeed, this is a very critical point. Prof. Penny Jeggo's group has shown that ATR phosphorylates ATM in hydroxyurea-treated cells (Stiff et al., EMBO J. 2006). In cells devoid of PR130 MS-275 cannot antagonize the hydroxyurea-evoked phosphorylation of ATM (**Fig. 6b**). This finding indicates that ATR activity is still intact in PR130 null cells that are incubated with MS-275. Accordingly, we see that ATR is not targeted by PR130 and MS-275 (**Fig. 1a** and **7a**).

Furthermore, we provide new details on the control of CHK1 phosphorylation by PR130. In our original manuscript, we speculated that the dephosphorylation of CHK1 in response to MS-275 relies on PP2A. Now we present a refined and more advanced model. We are able to distinguish two distinct novel molecular mechanisms. We demonstrate a direct dephosphorylation of pATM by PR130-PP2A and a cell cycle-dependent control of CHK1 phosphorylation that is modulated by PR130.

Our data show that in response to replicative stress, PR130 null cells progress slower from G1 to S phase than corresponding PR130 positive cells. However, the lower numbers of PR130 negative cells have even more replicative stress. They carry higher levels of RPA foci, pH2AX, pCHK1, and activated p53 together with a lower activity of WEE1 against CDK1 (**Figure 7a-d** and **8a-c**). We therefore conclude that PR130 is a novel regulator of the duration of this early and particularly replicative stress prone section of S phase.

MS-275 reduces both pCHK1 and WEE1. Accordingly, cell cycle control is lost and PR130 null cells that are arrested in early S phase disappear and the numbers of cells in G2/M phase and apoptosis increase (**Figure 8e-f**). We additionally show that increasing doses of hydroxyurea applied to HCT116 cells lead to an earlier arrest in S phase and to a dose-dependent increase in pCHK1 that does not require a corresponding increase in WEE1 activity (**Figure 7e-f**). These data suggest that an earlier arrest in S phase –evoked by a higher dose of hydroxyurea or a loss of PR130– leads to a more pronounced phosphorylation of CHK1. Also, our finding that the inhibition of PP2A with okadaic acid and cantharidin can rescue the hydroxyurea-induced phosphorylation of ATM, but not pCHK1 from dephosphorylation in the presence of MS-275 illustrates PP2A-dependent and -independent pathways (**Figure 5b** and **Supplementary Fig. S3a**).

The sustained activation of ATR in the presence of hydroxyurea plus MS-275 is verified by a phosphorylation of its target ATM in hydroxyurea plus MS-275 treated cells when its dephosphorylation is prevented by an elimination of PR130 (**Fig. 6b-c**). This ongoing activation of ATR can be explained by the ongoing replication in cells incubated with hydroxyurea and MS-275 (**Figs. 2d-g** and **3c-f**). The cells continue to replicate DNA and this unscheduled origin firing with the resulting delay in replication fork progression promotes the phosphorylation of ATR (and allows the phosphorylation of ATM by ATR if pATM is not dephosphorylated by the PR130-PP2A complex; **Fig. 6**). These findings agree with our observation that ATR catalyzes the phosphorylation of CHK1 in response to hydroxyurea (**Fig. 4d,f**) and that a rescue of ATM phosphorylation is therefore not able to increase the phosphorylation of CHK1 (**Fig. 5b**).

An alternative explanation for the results observed could be that, if unresolved replication forks are leading to double strand breaks in a time delayed manner leading to the activation of ATM/pCHK2 in a time-delayed manner, this may be sufficient to induce a WEE1 mediated G2/M arrest. The result suggests that HDAC1/2/3 mediated PR130 suppression may be activated following initial pATR/pCHK1 mediated replicative stress response to resolve any unrepaired breaks. pATM/pCHK2/WEE1 signaling may then be required for the subsequent G2/M arrest. This would be consistent with the authors observation that "MS-275 reduces HU-induced RPA foci (active ATR mediated intra-S phase checkpoints), while γ H2AX foci persist (Fig. 3c-d).

We thank the reviewer for making this important point and for sharing expertise. The reviewer's suggestion to analyze WEE1 has clearly advanced our work. The strong reduction of WEE1 by MS-275 (**Fig. 2f** and **Supplementary Fig. S4e**) provides an explanation why the S phase arrest cannot be maintained in cells treated with hydroxyurea and MS-275 (**Fig. 2a-g**). Furthermore, we offer data that we collected with the neutral comet assay, which detects dsDNA breaks. We demonstrate that ATM becomes phosphorylated in hydroxyurea-treated cells before the occurrence of such DNA lesions (**Supplementary Fig. S2b**). These data are consistent with the phosphorylation of ATM by ATR independent of dsDNA breaks (Stiff et al., EMBO J. 2006). In addition, a pre-incubation with MS-275 that suffices to induce PR130 can also suppress short-term hydroxyurea-induced checkpoint kinase signaling (**Fig. 5a**).

5. Furthermore, ATM inhibition increases HU-induced γ H2AX indicating DSBs (Fig. 4b). ATM is the kinase for phosphorylating H2AX at serine 139 and indeed the authors report that compared to isogenic wild-type cells, CHK2^{-/-} cells show less PARP1 cleavage and reduced γ H2AX indicative of a functional ATM/pCHK2 response to HU treatment (Fig. 4i). It is

unclear, as to how does its inhibition increase HU- induced DSB promoted H2AX phosphorylation. If the authors are invoking ATR as the kinase, they have to provide experimental evidence that in these cells if ATR is silenced by RNAi, H2AX phosphorylation is completely abrogated. Moreover, Fig 4b and 4d clearly shows that HU-induced pCHK1 phosphorylation is independent of ATM.

It has recently been shown that in response to hydroxyurea, ATR prevents the conversion of stress-induced single strand DNA into dsDNA breaks that are characterized by an increase of pH2AX (Toledo et al., Cell 2012). Our data confirm these published results (**Fig. 4f**). The mentioned article did not clarify if the ATR targets ATM and CHK1 are relevant to prevent DNA damage. Therefore, we thank the referee for pointing out that our new data suggest that ATM has a previously unknown protective effect on replication forks that are halted in the presence of hydroxyurea (**Fig. 4b**). In addition, we demonstrate such a function for CHK1 (**Fig. 4h**) and rule it out for CHK2 (**Fig. 4i**).

We were also puzzled by the observation that an inhibition of ATM augments the phosphorylation of H2AX while a deletion of CHK2 does not affect pH2AX initially and then attenuates the levels of pH2AX (**Fig. 4b,i**). Apparently, ATM and CHK2 have overlapping as well as divergent functions for the replicative stress response triggered by hydroxyurea. This finding is not unexpected, as CHK2 null cells also show impaired apoptosis and altered cell cycle control in response to gamma-irradiation. The fact that unlike ATM and p53 deleted mice, CHK2 null mice do not develop spontaneous tumors further shows that the ATM-CHK2 axis can be appreciated as a rather complex than linear signaling node (Hirao et al., MCB 2003).

We did not intend to state that ATR is the only kinase phosphorylating pH2AX. We are aware that there are for example reports showing that MAP kinases are responsible for the accumulation of pH2AX during stress (Köpfer et al., PNAS 2013). It is even impossible to test if ATR phosphorylates pH2AX in response to hydroxyurea, because the inhibition of ATR in cells treated with hydroxyurea exhausts the RPA pool covering replicative stress-induced single strand DNA and causes dsDNA breaks with high levels of pH2AX. Moreover, ATR inhibition disturbs the ATR-dependent regulation of the annealing helicase SMARCAL1 (Toledo et al., Cell 2012; Couch et al., Genes & Development 2013). Thus, inhibition of ATR leads to a breakdown of stalled replication forks and a double strand DNA break-dependent phosphorylation of H2AX and ATM (**Fig. 4f**). Due to these difficulties, we tried to alternatively address the question of the reviewer and we are happy to offer another set of experimental data. We treated HCT116 cells for 6 and 24 h with hydroxyurea and performed a neutral comet assay to measure dsDNA breaks. After a 6 h exposure to hydroxyurea, there are no comet tails in HCT116 cells. We only find them when we expose the cells for 24 h and with a

positive control (**Supplementary Fig. S2b**). Nevertheless, we see activation of ATM after 6 h and these findings are in perfect agreement with the phosphorylation of ATM by ATR in response to hydroxyurea (Stiff et al., EMBO J. 2006). Furthermore, we would like to mention that the increase in pATM after ATR inhibition in hydroxyurea-treated cells (**Fig. 4f**) is consistent with the well-established ATM activation in cells with dsDNA breaks. Furthermore, the reviewer pointed out that RNAi against ATR and its pharmacological inhibition illustrate an ATR-dependent phosphorylation of CHK1. These data are consistent with the literature (e.g., reviewed in Iliakis et al., Oncogene 2003). Concerning **Fig. 4d** and **4f**, we are grateful to the reviewer for mentioning that we verify CHK1 as a bona fide downstream target of ATR in hydroxyurea-treated cells. This finding entirely agrees with the notion that rescued pATM cannot promote the phosphorylation of CHK1 (**Fig. 5b**). We discuss the above-mentioned new data in the revised manuscript and we phrase more carefully.

6. Genetic experiments using PR130 deletion cell lines suggest that PR130 mediated regulation of replicative stress response cannot be attributed directly to changes in CHK1 phosphorylation. For example, while the HU- 210 induced phosphorylation of ATM remained elevated despite the presence of MS-275 (Fig. 6b). In contrast, MS-275 appears to decrease CHK1 phosphorylation in PR130 -negative cells and PR130 positive cells (Fig. 6c-d), which would also suggest that PR130 is dispensable for CHK1 phosphorylation, as in PR130 negative cells, there is no phosphatase subunit and CHK1 phosphorylation if it is PR130 dependent should be intact regardless of MS-275 addition!

Thus, two distinct molecular mechanism appear to be operational here. The paper suffers from the authors inability to separate these two events.

We are now able to fully separate these two events as stated in our response to Point 4. We show and discuss these new data in our revised manuscript.

Reviewer #3 (Remarks to the Author):

Goeder et al. investigated the role of histone deacetylases (HDACs) 1-3 in DNA replication checkpoint regulation, mostly in the epithelial colon cancer cell line HCT116. Long-term replication stress concomitant with HDAC inhibition decreased phosphorylation of the ATM and CHK1+2 kinases at key regulatory sites and this decreased checkpoint kinase phosphorylation correlated with checkpoint slippage and cell death. The decrease in phosphorylation pointed to the involvement of a phosphatase and indeed by a transcriptome profiling approach the authors identified a regulatory subunit of protein phosphatase 2A

(PP2A), PR130, as a potential candidate responsible for the checkpoint kinase dephosphorylation. Consistent with their hypothesis deletion of PR130 or siRNA driven knock-down of PR130 abrogated the decrease in ATM phosphorylation upon replication stress and HDAC inhibition. CHK1, on the other hand, still got dephosphorylated to the same extent as in wt PR130 cells suggesting the involvement of another phosphatase. However, the application of replication stress alone caused a significant increase of CHK1 phosphorylation in cells lacking PR130 still suggesting a role for PR130 in this process. Consistent with the enhanced CHK1 phosphorylation in the delta PR130 cells these cells arrested earlier than wt cells in G1/S phase. Upon replication stress these cells showed an increased number of RPA foci compared to wt cells and additional HDAC inhibition led to a decrease in RPA foci (indicating ssDNA) in both cell types. γ H2AX foci indicating double strand breaks were also increased in delta PR130 cells upon replication stress but additional HDAC inhibition did not further increase their level. Interestingly, more delta PR130 than wt cells underwent apoptosis (from the graph in Fig.7a it is difficult to read/deduce the % of cells in subG1) when exposed to replication stress and HDAC inhibition. The authors argued that the increased cell death is due to the attenuated CHK1 phosphorylation (ATM phosphorylation was unchanged) and the attenuated RAD51 levels.

In principle these are interesting findings in particular for the field of cancer therapy. A major weakness of this study is the lack of data on the mechanism underlying the observed phenomenon (PR130 regulating ATM and CHK1 phosphorylation). Moreover, induction of a 24h replication stress concomitant with a 24h inhibition of HDACs1-3, which are major epigenetic regulators of probably thousands of genes, is a very artificial condition. What is the physiological relevance of results made under such experimental conditions? What can we learn from these data about the “normal” replication checkpoint regulation by HDACs and PP2A? For example, the authors conclude from their data that PR130-PP2A “promotes dephosphorylation” of ATM and CHK1.

The authors should test directly if the PR130-PP2A holoenzyme is able to dephosphorylate ATM (at S1981, but also at S367 and S1893) and CHK1 (S317 and S345) and compare it to other di – or trimeric PP2A holoenzymes.

We thank the reviewer for the positive comments and as well for the critical assessment and the suggestions made. We now present a revised manuscript in which we present new details on the control of cell cycle progression by HDACs and PR130.

PR130 associates with PP2A A and C subunits (Creyghton et al., PNAS 2006 and Janssens et al., J Cell Sci. 2016). We correspondingly see that PR130 is relevant for the dephosphorylation of pATM in cells and that HDAC1 and HDAC2 suppress PR130 to

maintain the phosphorylation of ATM (**Figs. 5** and **6b-c**). ATM supports cell survival upon replicative stress (**Fig. 4a-c**). To corroborate these data and to follow the reviewer's suggestion, we immunoprecipitated PR130 complexes from cells treated with MS-275 and we tested their activity against a phosphorylated peptide around the S1981 of ATM. We present evidence that the PR130 complex is active against pATM (**Fig. 6f**). Moreover, this complex contains both the catalytic and the structural subunits of the PP2A complex (**Supplementary Fig. S4d**).

We would also be interested to see if hydroxyurea and MS-275 regulate the phosphorylation of ATM at S367 and S1893. Phosphorylation of these sites is induced by direct DNA damage due to gamma-irradiation (Kozlov et al., EMBO J 2006 and Kozlov et al., JBC 2011). We could only buy an antibody against ATM phosphorylated at S367 (ThermoFisher). When we applied this antibody to lysates from HCT116 cells that we had treated with hydroxyurea and MS-275, we noted no consistent changes in the phosphorylation of this site and no reproducible differences associated with the cellular PR130 status (n=3, representative example):

HCT116 ^{Δ gRNA} and HCT116 ^{Δ PR130} cells were treated with 2 μ M MS-275 and/or 1 mM hydroxyurea (HU) for 24 h. Indicated proteins and their phosphorylation were analyzed by Western blot; HSP90 as loading control. Please note that the lower panel is shown as Fig. 7d. The upper panel was made with the same lysates, i.e. both hydroxyurea and MS-275 worked well in this experiment.

We believe that our conditions of replicative stress do not alter these phosphorylation sites, which are induced by direct DNA damage. While there was no reason to analyze this further, we cite the more recent work by Kozlov and colleagues in light of the complexity of ATM phosphorylation sites.

Concerning pCHK1, Reviewer 3 believes that the higher phosphorylation of CHK1 in hydroxyurea-treated PR130 null cells relied on a direct interaction between PR130 and CHK1. We also believed this, but we found no interaction between CHK1 and PR130. Our new data suggest that an earlier arrest of cells in the G1/S transition causes this effect. This

finding is novel and shows that CHK1 is not only an upstream regulator of cell cycle progression, but also regulated by an alternative cell cycle regulation by PR130. Now we present a refined and more advanced model. We are able to distinguish two distinct novel molecular mechanisms. We demonstrate a direct dephosphorylation of pATM by PR130-PP2A and a cell cycle-dependent control of CHK1 phosphorylation that is modulated by PR130. Our data show that in response to replicative stress, PR130 null cells progress slower from G1 to S phase than corresponding PR130 positive cells. However, the lower numbers of cycling PR130 negative cells have even more replicative stress. They carry higher levels of RPA foci, pH2AX, pCHK1, and activated p53 together with a lower activity of WEE1 against CDK1 (**Figure 7a-d** and **8a-c**). We therefore conclude that PR130 is a novel regulator of this early section of S phase that is particularly prone to replicative stress. MS-275 reduces both pCHK1 and WEE1. Accordingly, cell cycle control is lost and PR130 null cells that are arrested in early S phase disappear and the numbers of cells in G2/M phase and apoptosis increase (**Figure 8e-f**).

We additionally show that increasing doses of hydroxyurea applied to HCT116 cells lead to an earlier arrest in S phase and to a dose-dependent increase in pCHK1 that does not require a corresponding increase in WEE1 activity (**Figure 7e-f**). These data suggest that an earlier arrest in S phase –evoked by a higher dose of hydroxyurea or a loss of PR130– leads to a more pronounced phosphorylation of CHK1.

Last but not least, our finding that the inhibition of PP2A with okadaic acid and cantharidin can rescue the hydroxyurea-induced phosphorylation of ATM, but not pCHK1 from dephosphorylation in the presence of MS-275 illustrates PP2A-dependent and -independent pathways (**Figure 5b** and **Supplementary Fig. S3a**). These data are also coherent with our observation that it is specifically the ATR kinase that catalyzes the phosphorylation of CHK1 in response to hydroxyurea (**Fig. 4d,f**) and that a rescue of ATM phosphorylation is therefore not able to increase the phosphorylation of CHK1 (**Fig. 5b**).

Regarding the other 17 B subunits, we analyzed our microarray for their expression. We found that MS-275 increased the mRNA levels of PR130/PPP2R3A and of B56 β /PPP2R5B (this factor was mentioned by the reviewer in her/his last comment), but no induction of other B subunits (**Supplementary Table 1**). Therefore, we tested the expression of this subunit at the protein level. Moreover, we also analyzed the levels of PR48/PPP2R3B, which was not increased in the array as a control. We found that B56 β and PR48 were not induced at the protein level by MS-275 and hydroxyurea plus MS-275 (**Supplementary Fig. S4c**). Thus, we conclude that the increase in PR130 is a restricted and specific effect of MS-275.

Moreover, in Fig. 6F an interaction is shown between ATM and PR130. Several questions arise from this result and have to be answered:

Is the fraction of ATM that is found in complex with PR130 dephosphorylated at the aforementioned sites?

We have shown in our initial submission that ATM being phosphorylated at S1981 can interact with PR130 (**Fig. 6d**; was 6F in the initial submission) and we can now demonstrate that HDAC activity prevents the interaction between PR130 and pATM (**Fig. 6d**). We had to carry out these experiments in the presence of okadaic acid as otherwise there would be hardly any pATM, which is the target of the PR130-PP2A complex. If hydroxyurea is given alone with okadaic acid, we cannot detect a complex between pATM and PR130 (**Fig. 6d**). This finding appears reasonable, because hydroxyurea can strongly promote the phosphorylation of ATM and a pre-existing binding of PR130-PP2A would prevent this posttranslational modification. We can further detect that PR130 is acetylated in cells and that MS-275 increases both its expression and even more pronouncedly its acetylation (**Fig. 6e**).

Our data showing that the PR130 complex from HCT116 cells is able to dephosphorylate the ATM phosphorylation site in a short peptide sequence demonstrate that such immunoprecipitates have dephosphorylating activity against pATM (**Fig. 6f**).

These data suggest that an increased expression and acetylation of PR130 in cells treated with the HDACi MS-275 promote the interaction of pATM with PR130-PP2A and the ensuing dephosphorylation of pATM by the catalytic subunit of the PP2A holoenzyme.

Is ATM also co-precipitating the other holoenzyme components (catalytic C and structural A subunit)? If yes, is the complex catalytically active when associated with ATM?

The question if the ATM-PP2A complex is catalytically active cannot be resolved due to a technical problem. If the PP2A is still bound to ATM, it cannot dephosphorylate another substrate. Moreover, the isolation of such a complex and testing its activity spans a time frame of at least 8 hours. It is not possible and not in accordance with a native catalytically active complex that it stays bound to its substrate for such long time. However, as discussed in the previous reply, we addressed the activity of the purified PR130 complex against a pATM sequence in a recombinant assay. This approach clearly shows that PR130 complexes are active against the phosphorylated sequence of ATM. Moreover, we demonstrate that PR130 complexes from MS-275 and MS-275 plus hydroxyurea treated cells contain the catalytically active and the structural subunit of PP2A and that HDAC inhibition by MS-275 augments the acetylation of PR130.

Is ATM only associated with PR130-holoenzymes upon replication stress and HDAC inhibition or is it already associated even in unstressed cells? If already bound what is the signal to get it activated?

We did not detect an interaction between PR130 and pATM in resting and hydroxyurea-treated cells. We now show that the interaction between PR130 and pATM is prevented by HDAC activity (**Fig. 6d**). Please see our response to Point 2 for a more detailed reply to this relevant question. In brief, our data suggest that an increased expression and acetylation of PR130 in cells treated with the HDACi MS-275 promote the interaction of pATM with PR130-PP2A and the ensuing dephosphorylation of pATM by the catalytic subunit of the PP2A holoenzyme.

The same analysis should be performed with CHK1 and PP2A.

According to our new data, CHK1 is not a direct target of PR130-PP2A. We explain this finding above in response to Point 1.

Minor points:

Not only a subset but all Western blot data need to be quantified (from at least 3 independent experiments), see below comments to Fig.5b and 5e.

We used the Odyssey System for most of our Western blots. This system allows the detection of antibody-based signals in a linear range and avoids problems with exponential signals and overexposure. Moreover, it allows the quantification of the signals directly. We quantified Fig. 5b and 5e as requested and all other Western blots which provide important information about a differential phosphorylation or expression. Due to the intense revision round our new manuscript contains a large number of new panels. None of the data represent a single experiment and all results were independently reproducible in various settings.

The authors should provide % for the cell cycle distribution graphs in Fig. 2a and 7a and should refer in the text to the exact numbers and phrase carefully. For example Fig.7c shows a decrease of RPA foci upon HDAC inhibition under replication stress, in delta PR130 cells there is a decrease from ~27 RPA foci/cell to ~22 and the authors wrote in line 230: "The MS-275-induced loss of RPA foci ...".

We provide this information in the text and we corrected the mistake as requested.

Line 178, the authors wrote that PP2A inhibition by cantharidin rescued the inhibition of checkpoint kinase phosphorylation. However, in Fig.5b there is still a substantial decrease in pATM and (even more) in pCHK1 phosphorylation levels visible despite the inhibition of PP2A. Therefore, as asked for above, all western blot data need to be quantified and secondly, the authors need to phrase more carefully their conclusions.

In our original manuscript, we speculated that the dephosphorylation of CHK1 in response to MS-275 relies on PP2A. Now we present a refined and more advanced model that we have briefly explained as response to the first issue raised.

Line 192 + Fig 5e, once again, the authors claim based on a single Western blot experiment that knockdown of HDAC1+2 would lead to increased expression of PR130. However, this is not evident from the single experiment shown in Fig.5e. This experiment needs to be repeated and the signals need to be quantified.

We are astonished that the reviewer mentioned that the experiment was carried out once only. We had done this experiment three times. In the revised manuscript, we state the numbers of independent experiments more explicitly. Furthermore, we have quantified all key experiments as requested and provide the signal intensities below the Western blot data. Moreover, we applied relevant statistical tests as indicated in the Material and Methods section.

Reviewers' Comments:

Reviewer #1:

Remarks to the Author:

the authors have done a great job in revising the ms

Reviewer #2:

Remarks to the Author:

Authors have addressed the concerns. The manuscript is now acceptable.

Reviewer #3:

Remarks to the Author:

The revised manuscript by Goeder et al. provides additional experimental evidence for their model that the protein phosphatase 2A (PP2A) regulatory subunit PR130 (PPP2R3A) directly or indirectly regulates the checkpoint kinases ATM1 and CHK1, respectively, and that HDAC1 and 2 sustain checkpoint kinase phosphorylation by suppressing the expression of PR130. The novel data reveal that HDAC1 and 2 bind to the PPP2R3A promoter and HDAC1/2 inhibition by MS-275 correlates with increased histone H3 acetylation at the PPP2R3A promoter. HDAC1/2 inhibition leads to increased expression (Fig. 5d, PR130 levels need to be quantified) and acetylation (New Fig. 6e) of PR130 and an interaction between PR130 and phosphorylated ATM (pATM) (Revised Fig. 6d). PP2A holoenzymes consisting of the PR130 subunit and the PP2A scaffold A and catalytic C subunits could be isolated from HDAC1/2 inhibited cells (Fig. S4d) and these holoenzymes dephosphorylated in vitro a phosphorylated peptide surrounding S1981 (New Fig. 6f). From this the authors concluded that HDAC1/2 inhibition triggers the interaction between pATM and PR130-PP2A holoenzymes and promotes directly the dephosphorylation of ATM. This major conclusion is not supported sufficiently by the additional data:

1) In Fig. 6d the authors show the interaction between pATM and PR130. Why did the authors not test the pATM-immunoprecipitate for the presence of the other components of the PP2A holoenzyme, the A and C subunit (as I had suggested in my previous review)? This is a key experiment (easy to do anyway) and needs to be done. If the other PP2A holoenzyme components are not associated with the pATM-PR130 then how (and when) are these subunits recruited into the complex to carry out the proposed function?

2) If we assume that PP2A holoenzyme is associated with ATM, is this PP2A complex catalytically active? The authors argue in their rebuttal that ATM bound PP2A cannot dephosphorylate another substrate. How do they know, have they done the experiment? Goodarzi et al. (EMBO J. 2004 Nov 10;23(22):4451-61) detected PP2A A and C subunit as well as PP2A activity in ATM immunoprecipitates. I wonder if the pS1981 ATM specific antibody, which the authors of the current study used for immunoprecipitating the pATM-PR130 complex, would anyway not impair PP2A's ability to dephosphorylate the pS1981 site. How does PP2A catalytic subunit gain access to the pS1981 with an antibody bound there? To be on the safe side I suggest doing the immunoprecipitation experiments also with a different ATM antibody. In the light of the aforementioned finding by Goodarzi et al., the authors need to check if in unstressed cells ATM interacts already with PP2A A and C subunits and PR130 might then only be recruited upon replicative stress.

3) Without comparing the catalytic activities of PR130-PP2A to other di-/trimeric PP2A holoenzymes towards the pS1981 peptide (this needs to be done at different substrate concentrations) the authors cannot conclude that PR130 is the targeting subunit for ATM dephosphorylation. This again is very important in the light of the results published by Goodarzi et al.

4) The PR130 knockdown and knockout experiments in HU+MS-275-treated HCT116 cells indicate that PR130 is necessary for the decreased pATM levels but is it also sufficient? E.g. will ectopic

expression of PR130 alone in HU-treated HCT116 cells lower pATM levels?

HDAC1/2 inhibition in HCT116 cells caused a cell cycle arrest in G1 phase and this correlated with reduced WEE1 and pCDK1 levels (new Fig 2f) but also with several fold increased levels of the CDK inhibitor p21 (new Fig. 2i). Neither in the result section nor the discussion did the authors comment on the increased p21 levels and the possible cell cycle consequences of the elevated p21 levels. The CDK inhibitor p21 is a known target of HDAC1 and 2 (Mol Cell Biol. 2010 Mar; 30(5):1171-81) and primary mouse fibroblasts lacking HDAC1 and 2 show a block in G1 phase that is associated with elevated p21 and p57(Kip2) levels (Genes Dev. 2010 Mar 1; 24(5):455-69). Interestingly, induction of replicative stress in the MS-275 treated cells caused a reduction of p21 levels (new Fig. 2i). The p21 levels should also be analyzed in the HCT116 cells lacking PR130 under the same conditions. Does PR130 presence/absence affect p21 regulation?

In HU treated cells lacking PR130 higher levels of phosphorylated pCHK1 are present than in PR130 wt cells. However, upon additional MS-275 treatment pCHK1 levels decreased to a similar extent in both cell types indicating that this dephosphorylation happens in a PR130 independent manner (Fig7a). Based on the "failure" of PP2A inhibitors to rescue the pCHK1 in HDAC-inhibited cells (Fig. 5b) the authors concluded that PP2A complexes are not regulating directly CHK1 phosphorylation. However, a closer look at Fig. 5b reveals that upon HU or HU+MS-275 treatment CHK1 phosphorylation increases 3-4 fold upon PP2A inhibition with 20 μ M Cantharidin. At 40 μ M Cantharidin pCHK1 levels stay high in HU-treated cells and in HU+MS-275 treated cells drop to ~ half of the levels in 20 μ M, which, however is still twice as much as in cells without PP2A inhibitor. In my opinion these data do not exclude the possibility that pCHK1 is a substrate of PP2A. The authors argue that CHK1 cannot be a direct target of PR130 because they could not detect a stable interaction between CHK1 and PR130 (data not shown). However, many PP2A-substrate interactions occur only transiently during catalysis and thus escape detection by standard methods. As I had suggested in my previous review, the authors should test in vitro if PR130 holoenzymes are able to dephosphorylate CHK1 (S317 and S345) and compare it to other di-/trimeric PP2A holoenzymes. Others have shown that PP2A is able to dephosphorylate directly CHK1 in vivo and in vitro (Mol Cell Biol. 2006 Oct; 26(20):7529-38) but these authors did not determine the specific PP2A holoenzyme responsible for CHK1 dephosphorylation. The evidence in the revised manuscript does not allow excluding the possibility of a direct PP2A-CHK1 substrate interaction.

The authors hypothesize that – based on lower levels of Wee1 in HU treated cells lacking PR130 (WEE1 levels in clone #16 are even increased) and the increased levels of pHDAC2 in these cells treated with MS-275 - that an earlier S phase arrest could explain the higher pCHK1 levels in cells without PR130. However, this is not evident from the Wee1 and pHDAC2 immunoblot signals in Fig. 7d. pCDK1 levels were quantified and seem to be decreased in HU-treated cells lacking PR130. Have the pCDK1 signals been normalized to the loading control levels (HSP90)? This is not indicated in the figure legend. In any case, without proper quantification such conclusions cannot be drawn. The Wee1 and pHDAC2 signals need to be quantified and normalized to the HSP90 control levels and dependent on the outcome the text has to be revised.

Minor points:

Fig. 5d, the increasing PR130 levels need to be quantified, the same accounts for the PR130 levels in HCT116 Δ gDNA cells in Fig.6b left panel.

Page 8, line 165: "compared to cells treated with alone" should read as "compared to cells treated with HU alone"

Throughout the discussion the authors refer too many times to specific figures, which makes the discussion read like a result section.

Point-by-point responses to all issues raised:

The reviewer stated that we provided additional experimental evidence for our model, but asked for new experiments and explanations. We provide further data and discuss our data regarding the literature and our novel results.

1. The reviewer asked in her/his comments #1-2:

#1: *"In Fig. 6d the authors show the interaction between pATM and PR130. Why did the authors not test the pATM-immunoprecipitate for the presence of the other components of the PP2A holoenzyme, the A and C subunit (as I had suggested in my previous review)? This is a key experiment (easy to do anyway) and needs to be done. If the other PP2A holoenzyme components are not associated with the pATM-PR130 then how (and when) are these subunits recruited into the complex to carry out the proposed function?"*

#2: *"If we assume that PP2A holoenzyme is associated with ATM, is this PP2A complex catalytically active? The authors argue in their rebuttal that ATM bound PP2A cannot dephosphorylate another substrate. How do they know, have they done the experiment? Goodarzi et al. (EMBO J. 2004 Nov 10;23(22):4451-61) detected PP2A A and C subunit as well as PP2A activity in ATM immunoprecipitates. I wonder if the pS1981 ATM specific antibody, which the authors of the current study used for immunoprecipitating the pATM-PR130 complex, would anyway not impair PP2A's ability to dephosphorylate the pS1981 site. How does PP2A catalytic subunit gain access to the pS1981 with an antibody bound there? To be on the safe side I suggest doing the immunoprecipitation experiments also with a different ATM antibody. In the light of the aforementioned finding by Goodarzi et al., the authors need to check if in unstressed cells ATM interacts already with PP2A A and C subunits and PR130 might then only be recruited upon replicative stress."*

These experimental suggestions are based on the work by Goodarzi and colleagues (*EMBO J.* 2004, Nov 10;23(22):4451-61) and we agree that the reviewer raised an interesting point. Following the reviewer's request, we ordered an antibody against exactly the epitope that was used in the publication by Goodarzi et al. (*EMBO J.* 2004 Nov 10;23(22):4451-61). These colleagues obtained the antibody from *Oncogene Research*, which was later on bought by *Millipore*, from which we received the antibody. We followed their immunoprecipitation protocol

precisely, but were unable to detect a basal interaction. Due to this negative experimental outcome, we systematically changed the parameters in the IP protocol, ranging from different antibody concentration to a fixation step with paraformaldehyde (PFA) to obtain the postulated complex. Below, we show a selection of experimental outcomes. Although we precipitated and enriched ATM with the antibody, we could never detect any PP2A-A or PP2A-C bound to immunoprecipitated ATM in lysates from HCT116 colon cancer cells and K562 leukemia cells:

Immunoprecipitation (IP) using 4 μ l (0.4 μ g) of ATM antibody (IgG, pre-immune IgG):

Immunoprecipitation using 8 μ l (0.8 μ g) of ATM antibody:

Immunoprecipitation using 40 μ l (4 μ g) of ATM antibody:

Immunoprecipitation using 8 μ l (0.8 μ g) of ATM antibody and PFA fixation:

Concerning the PFA experiment: ATM detection was performed with the same eluate on a second Western blot membrane, using only 15 μ l IP eluate compared to 60 μ l used for the lower blot; amounts of input were equal in both settings.

These findings do not match the Goodarzi publication in the *EMBO Journal*. This inconsistency could be due to the use of different cell lines. We included K562 leukemia cells, because leukemic cells were used in the Goodarzi paper. Also in K562 cells, there is no basal complex between PP2A and ATM. The inputs are clear positive controls in all experiments we provide. In our revised manuscript we refer to the data shown above as “data not shown”, but if the reviewer wishes, we will incorporate them as a supplementary figure.

We would like to stress that our data shown above are entirely consistent with the transient nature of PP2A-substrate interactions, a dephosphorylation reaction taking milliseconds, and a high turnover of substrate phosphorylation by PP2A. Thus, the interaction between ATM and PP2A could be too transient to be detectable. As the reviewer said (see below) “*However, many PP2A-substrate interactions occur only transiently during catalysis and thus escape detection by standard methods*”. We entirely agree with this remark. The hit-and-run mechanism exerted by PP2A is in perfect agreement with rapid dephosphorylation of PP2A substrates. Furthermore, we would like to point out that our genetic and pharmacological experiments unequivocally demonstrate that elimination of PR130 and inhibition of PP2A restores the phosphorylation of ATM in the presence of HDACi and PR130 (Figs. 5b, 6b-d, Supplementary Fig. 3a). We have clarified these issues in our revised manuscript.

Regarding additional experiments with the above-mentioned anti-ATM antibody, we noted that this antibody yields a non-specific band that is very close to PR130 in immunoprecipitations; please note the successful purification and the good enrichment of ATM in this IP. Hence, this antibody is useless for an analysis of a PR130/PP2A complex. Immunoblotting for PP2A-A and PP2A-C gave no positive signals (similar data were collected with another antibody against PP2A-C; not shown). Thus, there cannot be an enzymatic activity associated with such immunoprecipitations.

Immunoprecipitation of ATM from lysates of HCT116 cells that were treated with 2 μ M MS-275 or left untreated (Ctrl) for 24 h. IP was performed using 8 μ l (0.8 μ g) of ATM antibody (IgG, pre-immune IgG; Input = 2.5% of IP). This experiment particularly illustrates that it is mandatory to run an IP with pre-immune serum to avoid an erroneous identification of unspecific bands as PP2A-A/-C):

We obtained similar results with a second anti-ATM antibody from Abcam (ab32420). Another anti-ATM antibody from *Santa Cruz BT*, failed completely in both Western blot and immunoprecipitation (data not shown).

Maybe we should briefly note that evidence against a role of PP2A for the phosphorylation of ATM in unstressed cells has been collected in two reports using *Xenopus* extracts (Petersen et al., *Mol. Cell Biol.* 2006 and You et al., *Nature Cell Biology* 2007). Furthermore, several groups have immunoprecipitated ATM from cell extracts and could induce its

autophosphorylation without the need to inhibit PP2A (e.g., Kozlov et al., *J. Biol. Chemistry* 2003; Lee and Paull, *Science* 2005).

The reviewer asked whether PR130-PP2A gets recruited to ATM in response to replicative stress. However, according to our data, the recruitment of PR130, which binds both PP2A-A/C and accumulates upon HDAC inhibition, leads to a dephosphorylation of pATM. We show this in several experiments, in which we used HDACi, knocked down HDAC1/HDAC2, and eliminated PR130 (Figs. 1, Supplementary Fig. 1, Fig. 5c-h; 6b-f).

We provide the following figure to depict our new data and we hope this makes it easier to understand our reasoning.

An increase in PR130 upon HDAC inhibition or a knockdown of HDAC1/HDAC2 leads to a dephosphorylation of ATM phosphorylation in response to hydroxyurea. These data reveal a novel mechanistic link between the epigenetic modifiers HDAC1 and HDAC2, PR130, and checkpoint kinase signaling controlling key cell fate decisions.

The finding that post-translational modifications regulate B-type subunits of PP2A and thereby recruit PP2A-A/C is also seen in other systems. For example, protein kinase A phosphorylates the B56 δ subunit to direct PP2A activity in neurons (Ahn et al., *PNAS* 2006). Our work is the first that links PR130 to checkpoint kinase signaling. HDAC1 and HDAC2 bind to the *PR130* promoter *in vivo* and there is increased expression of PR130 and increased acetylation of PR130 when HDAC1 and HDAC2 are inhibited by the HDACi MS-275. As a consequence, phosphorylated ATM becomes dephosphorylated (Figs. 5c-h, 6b-f). ATM phosphorylation alone does not trigger its interaction with PR130 (Fig. 6d). This is logical, as there would

otherwise be no possibility to phosphorylate ATM. We improved the presentation of our findings in the revised manuscript.

2. The reviewer then mentioned: *Without comparing the catalytic activities of PR130-PP2A to other di-/trimeric PP2A holoenzymes towards the pS1981 peptide (this needs to be done at different substrate concentrations) the authors cannot conclude that PR130 is the targeting subunit for ATM dephosphorylation. This again is very important in the light of the results published by Goodarzi et al.*

We show that the PR130-PP2A complex can dephosphorylate this peptide (the peptide concentration was chosen according to the instructions by Millipore, which provided the positive control at this concentration). Different substrate concentrations will not yield any additional information or value for the overall conclusions of the manuscript.

In our revised manuscript we demonstrate that PR130, but not other B-type subunits, show an increased expression after HDAC inhibition (Figs. 5c-g, Supplementary Fig. 4d, Supplementary Table 1). Most importantly, elimination of PR130 by two independent genetic techniques (RNAi and CRISPR-Cas9), which is a gold standard in our view, ablates the negative effects of HDACi on ATM phosphorylation (Figs. 6b-c). Furthermore, we demonstrate that HDAC inhibition with MS-275 induces PR130, which can bind PP2A-A and PP2A-C (Figs. 6d-f). Therefore, we can conclude that in our experimental context and cellular setting, PR130 is the targeting subunit. In combination with several new mechanisms that we elaborated, this is the main and novel conclusion of our manuscript. We agree that under different conditions other subunits also target ATM. For example, the B55 subunit can dephosphorylate ATM in cells infected with Adenovirus (Brestovitsky and colleagues, *PLOS Pathogens* 2016) and hence there will be a dephosphorylating activity associated with B55 when it is used in an *in vitro* setting. This is in perfect line with the complexity of checkpoint signaling and its diverse roles in biology. However, experimentally determining a further context in which other subunits target ATM is by far beyond the scope of the current manuscript and will not add significant information to the new mechanisms we describe. Hence, it makes no sense to eliminate other B-type subunits. Moreover, the elimination of all 17 B-type subunits (alone and in combination) would take several years and the elimination of some B-type subunits impairs cell growth, which poses an obstacle (see e.g., Sablina and colleagues, *Cancer Res.* 2010).

Regarding dimeric complexes: It is not possible to isolate PP2A-A and PP2A-C without their attachment to B-type subunits. There is no covalent interaction between PP2A-A/C and harsh immunoprecipitation conditions will also break up their interaction. Thus, this experiment cannot be carried out technically. Establishing a fully recombinant biochemical assay for PP2A-A/C and the at least 17 subunits would again take several years. It might even be impossible, as to be functional this trimeric complex requires posttranslational modifications from cells, which are not even characterized in detail. Furthermore, the stoichiometry of PP2A-A/C complexes that are optimal for checkpoint kinase dephosphorylation are not known. Clearly, such questions are rather suited for a specialized biochemical journal and far beyond the scope of our work.

3. *The PR130 knockdown and knockout experiments in HU+MS-275-treated HCT116 cells indicate that PR130 is necessary for the decreased pATM levels but is it also sufficient? E.g. will ectopic expression of PR130 alone in HU-treated HCT116 cells lower pATM levels?*

We did the experiment as requested (n=3, all measurements done). Its outcome perfectly confirms our model, in which PR130 levels regulate the dephosphorylation of ATM by PP2A. Thus, there is mass action of PR130 on pATM. Our observation that the phosphorylation of CHK1 was unaffected additionally supports our hypothesis that PR130 targets pATM but not pCHK1 (compare to Figs. 6b and 7a). We show a typical experimental outcome, which we include in the revised manuscript (Fig. 5h).

Numbers indicate densitometric analysis of Western blot signals normalized to their respective loading controls (mSIN3A, Vinculin) and relative to hydroxyurea (HU) treated cells.

HDAC1/2 inhibition in HCT116 cells caused a cell cycle arrest in G1 phase and this correlated with reduced WEE1 and pCDK1 levels (new Fig 2f) but also with several fold increased levels of the CDK inhibitor p21 (new Fig. 2i). Neither in the result section nor the discussion did the authors comment on the increased p21 levels and the possible cell cycle consequences of the elevated p21 levels. The CDK inhibitor p21 is a known target of HDAC1 and 2 (Mol Cell Biol. 2010 Mar; 30(5):1171-81) and primary mouse fibroblasts lacking HDAC1 and 2 show a block in G1 phase that is associated with elevated p21 and p57(Kip2) levels (Genes Dev. 2010 Mar 1; 24(5):455-69). Interestingly, induction of replicative stress in the MS-275 treated cells caused a reduction of p21 levels (new Fig. 2i). The p21 levels should also be analyzed in the HCT116 cells lacking PR130 under the same conditions. Does PR130 presence/absence affect p21 regulation?

We agree with the reviewer that the reduction of the CDK inhibitor p21 in cells treated with hydroxyurea and MS-275 (Figs. 2i and 7e) is an interesting finding and we are grateful for the good suggestions.

Hydroxyurea induces p53 and its target gene p21 in HCT116 cells (Beckermann et al., *Genes Dev.* 2009). The induction of p21 by MS-275 does not require p53 in HCT116 cells (Sonnemann et al., *Br. J. Cancer.* 2014). MS-275 attenuates the hydroxyurea-induced p53 phosphorylation, expression, and activity and this also affects the p53 target gene p21 in HCT116 cells (Fig. 2h-i). This observation, together with reduced WEE1/pCDK1 and remaining Cyclin B1 (Fig. 2f-g) agrees entirely with the entry of cells into mitotic catastrophe (Fig. 2a-e). Interestingly, PR130 affects the induction of p21 and the phosphorylation-dependent activation of its upstream regulator p53 (Figs. 6b and 7e). We also noted that PR130 null cells have more p21 than cells with PR130 (Fig. 7e). This finding provides a nice explanation for the early G1/S phase arrest of PR130 null cells when they are exposed to hydroxyurea (Fig. 7c). MS-275 also induces p21, but the hydroxyurea/MS-275 combination decreases p21 levels (Fig. 7c) and this is again coherent with the loss of the hydroxyurea-induced cell cycle arrest. The reduction of pCDK1 in cells lacking PR130 is a further explanation for the loss of the

hydroxyurea-induced, early G1/S phase arrest when HDAC1/HDAC2 are inhibited by MS-275 (Fig. 7c-d).

Therefore, we asked whether the earlier G1 arrest in cells treated with increasing doses of hydroxyurea (Fig. 7f) is also linked to p21. Indeed, a proportional increase in p21 is associated with an earlier G1 arrest in HCT116 cells (Fig. 7g). This figure also illustrates that a dose-dependent increase in the levels of p21, but not of pCDK1, correlates with an earlier G1 phase arrest and higher levels of pCHK1 in response to hydroxyurea (Fig. 7f-g).

We have added the citations that the reviewer mentioned and discuss the new data on p21.

- In HU treated cells lacking PR130 higher levels of phosphorylated pCHK1 are present than in PR130 wt cells. However, upon additional MS-275 treatment pCHK1 levels decreased to a similar extent in both cell types indicating that this dephosphorylation happens in a PR130 independent manner (Fig7a). Based on the “failure” of PP2A inhibitors to rescue the pCHK1 in HDAC-inhibited cells (Fig. 5b) the authors concluded that PP2A complexes are not regulating directly CHK1 phosphorylation. However, a closer look at Fig. 5b reveals that upon HU or HU+MS-275 treatment CHK1 phosphorylation increases 3-4 fold upon PP2A inhibition with 20 μ M Cantharidin. At 40 μ M Cantharidin pCHK1 levels stay high in HU-treated cells and in HU+MS-275 treated cells drop to ~ half of the levels in 20 μ M, which, however is still twice as much as in cells without PP2A inhibitor. In my opinion these data do not exclude the possibility that pCHK1 is a substrate of PP2A. The authors argue that CHK1 cannot be a direct target of PR130 because they could not detect a stable interaction between CHK1 and PR130 (data not shown). However, many PP2A-substrate interactions occur only transiently during catalysis and thus escape detection by standard methods. As I had suggested in my previous review, the authors should test in vitro if PR130 holoenzymes are able to dephosphorylate CHK1 (S317 and S345) and compare it to other di-/trimeric PP2A holoenzymes. Others have shown that PP2A is able to dephosphorylate directly CHK1 in vivo and in vitro (Mol Cell Biol. 2006 Oct; 26(20):7529-38) but these authors did not determine the specific PP2A holoenzyme responsible for CHK1 dephosphorylation. The evidence in the revised manuscript does not allow excluding the possibility of a direct PP2A-CHK1 substrate interaction.*

We entirely agree that PP2A can target pCHK1. We corrected the manuscript as requested, and we cite the mentioned reference. In our experimental setting, MS-275 reduces the

phosphorylation of ATM dependent on the activity of PP2A. The PP2A inhibitors cantharidin and okadaic acid restore the phosphorylation of ATM. Cantharidin also has an effect on the MS-275-mediated reduction of CHK1 phosphorylation by hydroxyurea, but okadaic acid shows only a very mild effect in this setting (Fig. 5b and Supplementary Fig. 3a). Since okadaic acid is far more selective than cantharidin for PP2A over PP1 (McCluskey et al., *Bioorganic&Medicinal Chemistry Letters* 2002: Okadaic acid PP1 IC₅₀=60nM, PP2A IC₅₀=1nM; cantharidin PP1 IC₅₀=0.16μM, PP2A IC₅₀=1.7μM), we moved the data collected with this agent into the main figures and placed the data collected with cantharidin in the supplemental section (Fig. 5b and Supplementary Fig. 3a). This decision is also consistent with the use of okadaic acid to block PP2A in Fig. 6d.

In our presented work, we focus on PR130, as we see that this B-type subunit accumulates upon an inhibition of HDAC1/HDAC2. Most importantly, elimination and overexpression of PR130 show that PR130 levels are crucial for the phosphorylation of ATM, but not of CHK1, *in vivo* (Figs. 5h, 6b-c, 7a-b). Hence, even if a PR130-PP2A complex dephosphorylates pCHK1 *in vitro*, this outcome would not reflect the cellular reactions. Such a finding would rather be an *in vitro* artifact that does not add value to our work.

Our new data show that PR130 integrates the control of CHK1 phosphorylation by cell cycle regulatory molecules (p53/p21 and WEE1/pCDK1 signaling nodes; Fig. 7a-g). Such a finding is coherent with the literature, which states that cell cycle progression regulates the phosphorylation of CHK1 (Dobbelstein and Sørensen, *Nat Rev Drug Discov* 2015; Mahajan and Mahajan, *Trends Genet.* 2013; Saini et al., *Oncotarget* 2015; Xu N, et al. *Oncogene.* 2012).

As said above for dimeric complexes: It is not possible to isolate PP2A-A and PP2A-C from cells without having them attached to B-type subunits. Moreover, unpredictable issues on posttranslational modifications and stoichiometry may prevent or cause artifacts in this analysis.

The authors hypothesize that – based on lower levels of Wee1 in HU treated cells lacking PR130 (WEE1 levels in clone #16 are even increased) and the increased levels of pHDAC2 in these cells treated with MS-275 - that an earlier S phase arrest could explain the higher pCHK1 levels in cells without PR130. However, this is not evident from the Wee1 and pHDAC2 immunoblot signals in Fig. 7d. pCDK1 levels were quantified and seem to be decreased in HU-treated cells lacking PR130. Have the pCDK1 signals been normalized to the loading control levels (HSP90)? This is not

indicated in the figure legend. In any case, without proper quantification such conclusions cannot be drawn. The Wee1 and pHDAC2 signals need to be quantified and normalized to the HSP90 control levels and dependent on the outcome the text has to be revised.

We agree that showing the measurements adds value to figure 7d. As we stated in the figure legend, we made the measurements (signals normalized to HSP90).

We did not imply that the phosphorylation of HDAC2 is linked to cell cycle arrest or the phosphorylation of CHK1. HDAC2 phosphorylation is not consistently affected by the PR130 knockout (this question was originally asked by Reviewer 2; there is only a non-significant trend towards increased phosphorylation of HDAC2 by MS-275). Concerning WEE1, we find that its activity rather than its level depend on PR130. WEE1 phosphorylates CDK1 to slow cell cycle progression (Mahajan and Mahajan, *Trends Genet.* 2013). The inhibition of WEE1/pCDK1 by MS-275 in hydroxyurea-treated cells -together with the above-mentioned regulation of p21- provides an explanation for the loss of cell cycle control (Figs. 2 and 7c-g).

5. *Minor points:*

Fig. 5d, the increasing PR130 levels need to be quantified, the same accounts for the PR130 levels in HCT116ΔgDNA cells in Fig.6b left panel.

Page 8, line 165: “compared to cells treated with alone” should read as “compared to cells treated with HU alone”

Throughout the discussion the authors refer too many times to specific figures, which makes the discussion read like a result section.

We did the quantifications (Supplementary Fig. 4b) including statistics and corrected the minor mistakes. The references to the figures were intended to improve the readability of our complex work with now over 80 panels. Nevertheless, we have removed the references to the figures as suggested by the reviewer.

Reviewers' Comments:

Reviewer #3:

Remarks to the Author:

The authors have addressed all the concerns.

REVIEWER'S COMMENTS

Reviewer #3 (Remarks to the Author):

The authors have addressed all the concerns.

We thank the reviewer for the positive assessment of our work.